



# Effects of finite source rupture on landslide triggering:
# The 2016 $M_W$ 7.1 Kumamoto earthquake

Sebastian von Specht[1,2], Ugur Ozturk[1,2,3], Georg Veh[1], Fabrice Cotton[2,1], and Oliver Korup[1]

[1]University of Potsdam, Institute of Earth and Environmental Science, Karl-Liebknecht-Str. 24-25, 14476 Potsdam-Golm, Germany
[2]Helmholtz Centre Potsdam - GFZ German Research Centre for Geosciences, Telegrafenberg, 14473 Potsdam, Germany
[3]Potsdam Institute for Climate Impact Research (PIK) e. V., Telegrafenberg, 14473 Potsdam, Germany

**Correspondence:** Sebastian von Specht (sspecht@uni-potsdam.de)

**Abstract.** The propagation of a seismic rupture on a fault introduces spatial variations in the seismic wavefield surrounding the fault during an earthquake. This directivity effect results in larger shaking amplitudes in the rupture propagation direction. Its seismic radiation pattern also causes amplitude variations between the strike-normal and strike-parallel components of horizontal ground motion. We investigated the landslide response to these effects during the 2016 Kumamoto earthquake ($M_W$ 7.1) in central Kyūshū (Japan). Although the distribution of some 1,500 earthquake-triggered landslides as function of rupture distance is consistent with the observed Arias intensity, the landslides are more concentrated to the northeast of the southwest-northeast striking rupture. We examined several landslide susceptibility factors: hillslope inclination, median amplification factor (MAF) of ground shaking, lithology, land cover, and topographic wetness. None of these factors can sufficiently explain the landslide distribution or orientation (aspect), although the landslide headscarps coincide with elevated hillslope inclination and MAF. We propose a new physics-based ground motion model that accounts for the seismic rupture effects, and demonstrate that the low-frequency seismic radiation pattern consistent with the overall landslide distribution. The spatial landslide distribution is primarily influenced by the rupture directivity effect, whereas landslide aspect is influenced by amplitude variations between the fault-normal and fault-parallel motion at frequencies $< 2$ Hz. This azimuth-dependence implies that comparable landslide concentrations can occur at different distances from the rupture. This quantitative link between the prevalent landslide aspect and the low-frequency seismic radiation pattern can improve coseismic landslide hazard assessment.

## 1 Introduction

Landslides are one of the most obvious and hazardous consequences of earthquakes. Acceleration of seismic waves alters the force balance in hillslopes and temporarily exceed cohesion and friction (Newmark, 1965; Dang et al., 2016). Increased landslide rates have been reported on hillslopes close to earthquake rupture, mostly tied to ground acceleration (Gorum et al., 2011) and lithology (Chigira and Yagi, 2006). Substantial geomorphological and seismological data sets are required to assess the response of landslides to ground motion and a growing number of studies has shed light on the underlying links (e.g. Lee, 2013; Allstadt et al., 2018; Roback et al., 2018). Several seismic measures such as vertical and horizontal peak ground acceleration (PGA) (Miles and Keefer, 2009), root-mean square (RMS) acceleration or Arias intensity ($I_A$) (Arias, 1970; Keefer,





**Figure 1.** The area of Kyūshū affected by coseismic landslides triggered by the 2016 $M_W$ 7.1 Kumamoto earthquake. The colored patch is the slip distribution of the rupture model of Kubo et al. (2016), the dashed box includes landslides related to the triggered event in Yufu. The inset map shows the station network within 150 km of the rupture.



1984; Harp and Wilson, 1995; Jibson et al., 2000; Jibson, 2007; Torgoev and Havenith, 2016), seismic source-moment release, hypocentral depth, as well as rupture extent and propagation (Newmark, 1965) correlate with landslide density (Meunier et al., 2007).

Landslides concentrate in the area of strongest ground acceleration (Meunier et al., 2007), whereas total landslide area decreases from the earthquake rupture with the attenuation of peak ground acceleration (Dadson et al., 2004; Taylor et al., 1986). Those metrics are combined with morphometrics (e.g. local hillslope inclination, curvature) and geological parameters (e.g. lithology, geological structure, land cover) (Gorum et al., 2011; Havenith et al., 2015), since they could alter landslide susceptibility (Pawluszek and Borkowski, 2017), in addition to seismic amplification (Maufroy et al., 2015). For instance, Tang et al. (2018) found that lithology, PGA, and distance from the rupture plane are important in assessing coseismic landslide distribution that were triggered by the 2008 Wenchuan earthquake ($M_W$ 7.9).

On April 16, 2016 at 16:25 UTC central Kyūshū was hit by a $M_W$ 7.1 earthquake (Fig. 1). The left-lateral dip-slip event ruptured along the Futagawa and Hinagu faults striking NW-SE with a hypocentral depth of 11 km (e.g. Kubo et al., 2016). The rupture propagated northeastward and stopped at Mt. Aso. Fault source inversion studies show a northeast propagation of the rupture originating under Kumamoto city with highest slip near the surface at the western rim of the Aso caldera (e.g. Kubo et al., 2016; Asano and Iwata, 2016; Moore et al., 2017; Uchide et al., 2016; Yagi et al., 2016; Yoshida et al., 2017). The earthquake triggered approximately 1,500 landslides (National Research Institute for Earth Science and Disaster Prevention, 2016) that concentrated mainly inside the caldera and the flanks of Mt. Aso on the Pleistocene and Holocene lava flow deposits (Paudel et al., 2008; Sidle and Chigira, 2004), although most of the terrain near the earthquake rupture is rugged (Fig. 1). Thus, we hypothesize that rupture directivity causes an asymmetric distribution of landslides around the rupture plane, because of more severe ground motion along the propagating rupture (Somerville et al., 1997). Similarly asymmetric landslide distributions attributed to rupture directivity were repoprted for the 2002 Denali earthquake ($M_W$ 7.9) (Frankel, 2004; Gorum et al., 2014), and the 2015 Gorkha earthquake ($M_W$ 7.8) (Roback et al., 2018). In case of the 1999 Chi-Chi earthquake, Lee (2013) speculated that the prevalent landslide aspects were correlated to the fault movement direction (Ji et al., 2003; Meunier et al., 2008). These observations indicate that the rupture process introduces variations on the incoming energy on hillslopes.

Here we link those dominant near-surface seismic characteristics relevant to the pattern and orientation of coseismic landslides. To this end we investigate the geological conditions (lithology, aspect, hillslope inclination, topographic amplification, soil wetness) in central Kyūshū as well as seismic waveform records from 240 seismic stations within 150 km of the rupture (Fig. 1). The two most prominent seismic effects—well founded in seismological theory (e.g. Aki and Richards, 2002) and documented in empirical relationships (e.g. Somerville et al., 1997)—are the rupture directivity and amplitude variations of fault-normal and fault-parallel motion. We examine whether the geomorphic characteristics around the Aso caldera made this area more susceptible to landslides than the surrounding topography near the earthquake rupture; or instead rupture effects control the asymmetric distribution of the landslides. We introduce a ground motion metric related to azimuth-dependent seismic energy (i.e. related to seismic velocity), because these effects attenuate with increasing frequency and are less captured by acceleration based metrics. From this we propose a new ground-motion model that is consistent with the observed coseismic landslide pattern.





## 2 Data

We combine data sets on the response of landslides to the earthquake, including topography, land cover, geology, seismic waveforms, velocity structure, near surface characteristics, and landslide location and planform.

### 2.1 Topographic data

Most topographic data used in this study are provided by the Japan Aerospace Exploration Agency (JAXA) and its Advanced Land Observing Satellite (ALOS) project with a horizontal resolution of 1" ($\approx 30$ m). This digital surface model (DSM) forms the basis for computing aspect, hillslope inclination (Fig. 2g), the median amplification factor (MAF Maufroy et al., 2015) (Fig. 2h), and the topographic wetness index (Böhner and Selige, 2006) (Fig. 2i). The ALOS project also provides data on land cover including anthropogenic influence (sealing, agriculture) and vegetation (Fig. 2d). While data on major geological units are from the Seamless Digital Geological Map of Japan by the Geological Survey of Japan (Fig. 2a).

### 2.2 Topographic amplification of ground motion

Topographic features, such as mountains and valleys, can amplify or attenuate seismic waves (Massa et al., 2014; Maufroy et al., 2012, 2015). Largest ground-motion variations occur on hillslopes and summits, whereas variations are intermediate on narrow ridges, and low on valley floors. Maufroy et al. (2015) introduced proxies for these topographic site effects, of which we use the median amplification factor (MAF), based on the topographic curvature, and the S-wave velocity $v_S$ traveling at frequency $f$:

$$MAF(f) = 8 \times 10^{-4} \frac{v_S}{f} C_S \left( \frac{v_S}{2f} \right) + 1 \tag{1}$$

where $C_S \left( \frac{v_S}{2f} \right)$ is the topographic curvature convolved with a normalized smoothing kernel based on two 2D box-car functions as a function of $v_S$ and $f$.

The curvature is estimated from the DSM (Zevenbergen and Thorne, 1987; Maufroy et al., 2015) and the seismic velocity $v_S$ is the average S-wave velocity of the uppermost 500 m from the model of Koketsu et al. (2012). The frequency $f$ of the seismic wave is the fundamental frequency of the hillslope section on which landsliding occurred (Massa et al., 2014).

Another site effect that influences landslide potential is the local soil or groundwater content, which can be modeled for uniform conditions to first order using the topographic wetness index (TWI) of Böhner and Selige (2006):

$$TWI = \log \frac{A_c}{\tan \beta}, \tag{2}$$

where $A_c$ is the upslope catchment area and $\beta$ is the hillslope inclination derived from the DSM with filled sinks (Planchon and Darboux, 2001).





**Figure 2.** a) Geology of central Kyūshū. The most common geological units of the landslides (black dots) are shown in b). For the landslide affected area (outer black line) the dominant geological units are in c). The inner black line denotes the rupture area, containing the hypocenter (black diamond). d) Land cover. Land cover in the landslide areas is shown in e) and for the the entire landslide affected area in f). g) Hillslope inclination. h) Median amplification factor (MAF). i) Topographic Wetness Index (TWI).



### 2.3 Ground motion data

Ground-motion data are from the Kik-Net/K-Net of the National Research Institute for Earth Science and Disaster Prevention (NIED) of Japan. NIED operates for Kik-Net both borehole and surface stations, and we use the latter only. The Japan Meteorological Agency (JMA) also released seismic data from the municipal seismic network for the largest earthquakes of the Kumamoto sequence. In total, data from 240 stations in Kyūshū are available with complete azimuthal coverage within 150 km from the earthquake rupture (Fig. 1).

The analysis of seismic waveforms is based on accelerometric data only. Both the NIED and JMA data are unprocessed and we follow the strong motion processing guidelines by Boore and Bommer (2005). We use both acceleration and velocity in further processing, and integrate the accelerograms to obtain velocity records. We correct the data with the automated baseline correction routine by Wang et al. (2011). The JMA accelerometric data further require a piece-wise baseline correction prior to the displacement baseline correction due to abrupt (possibly instrument related) jumps (Boore and Bommer, 2005; Yamada et al., 2007). We use the automated correction for baseline jumps by (von Specht, 2018 [in. prep.]).

An earthquake was triggered approximatley 80 km to the northeast in Yufu 32 s after the Kumamoto earthquake (Uchide et al., 2016) (Fig. 1). Due to the close succession of the two events, waveforms of the triggered event interfere with the coda of the Kumamoto earthquake. We taper the data to reduce signal contributions by the triggered event. The taper position is based on theoretical traveltime differences between the P wave ($v_P = 5700$ m s$^{-1}$) arrival of the Kumamoto earthquake and the S wave arrival ($v_S = 3300$ m s$^{-1}$) of the triggered event. The respective travel paths to the stations are measured from the hypocenters. Since fewer instruments are located to the northeast and the triggered event lose to the sea, less than 10 % of the data are strongly contaminated by the triggered event.

NIED hosts the rupture plane model of Kubo et al. (2016), which describes the slip history on a curved rupture plane (based on the surface traces of the Futagawa and Hinagu faults) with a total length of 53.5 km and 24.0 km width (Fig. 1). We use the extent and shape of the rupture plane to estimate the landslide affected area and to define the rupture plane distance $r_{rup}$, the shortest distance from the rupture plane.

The underground structure in terms of seismic velocities ($v_P$, $v_S$) and density ($\rho$) (Koketsu et al., 2012) are available for 23 layers down to the mantle in $\approx 0.1$ degree resolution covering all of Japan; we only consider the layers of the upper 0.5 km to compute a velocity average for the computation of MAF.

NIED provides data for the subsurface shear wave velocities ($v_{S30}$) as well as site amplifications factors $S_{amp}$. Contrary to $v_S$ from Koketsu et al. (2012), $v_{S30}$ is derived for the upper 30 m only and more suitable for energy estimates, which require velocities at the surface (recording station). The site amplification factor $S_{amp}$ describes by how much seismic waves are amplified independent of their frequency.

### 2.4 Landslide data

Detailed landslide data are provided by NIED as polygons (Fig. 1), mapped from aerial imagery at different times after the Kumamoto earthquake. The first data set contains landslides that were identified between April 16 to 20, though the area close




to the summit of Mt. Aso was not covered. A second data set was collected on April 29, 2016 and covers those parts of Mt. Aso that remained unmapped. However, the second data set may contain rainfall induced landslides, since the rainy season in Kyūshū starts in May (Matsumoto, 1989), and there was rainfall after the Kumamoto earthquake and landslides triggered by volcanic activity. We selectively combine the two data sets for this study, using only those landslides from the second

database, which are also partly present in the first data set. We exclude any rainfall triggered landslides with this approach, though possibly omitting seismically induced landslides exclusive to the second database. However, the area in question is comparatively small to the full extent of the study area, and the missing landslides are considered to be minor in terms of their area.

Several landslides cluster ≈ 80 km to the northeast of the mainshock in the municipalities Yufu and Beppu (Fig. 1), As

mentioned above, this area was hit by a triggered earthquake (Uchide et al., 2016). We believe that the distant northeastern landslides were induced by this triggered event. This also explains the considerable gap of landslides (≈ 50 km) between Yufu and Aso (Fig. 1) in otherwise steep topography. Hence we exclude the landslides from Yufu and Beppu ($< 10\%$ of all landslides, $< 3\%$ of total landslide area) from our database.

Apart from the special release of landslide data for the 2016 Kumamoto Earthquake, NIED hosts a landslide database for

all Japan (National Research Institute for Earth Science and Disaster Prevention, 2014). This database covers unspecified landslides of any origin. We extract a subset from this landslide database to compare it with the landslides triggered by the Kumamoto earthquake. Contrary to the special Kumamoto release, only the landslide deposits are given as polygons, whereas the scarps are mapped as lines. We manually define polygons representing the total landslide area bound by the scarp line and covering the deposit area to make both data sets comparable. This step is essential, because the landslide source area is

generally not identical with the deposit area.

## 3  Total area affected by landsliding

We define the landslide affected area, in which coseismic landsliding occurred, as the area spanned by the rupture plane distance covering 97.5 % of the total landslide area (Harp and Wilson, 1995; Marc et al., 2017). Thus the total landslide affected area is 3968.6 km² and within 22.9 km distance from the rupture plane.

An $M_W$ 7.1 event with a fault length of 53.5 km and an asperity length of 12.78 km (3 km) results in a landslide affected area of 3914 km² (4406 km²) using parameters proposed Marc et al. (2016). We derived the event depth of 11.1 km as the moment weighted average of the rupture model of Kubo et al. (2016). Both estimates are consistent with our area estimate. Marc et al. (2016) introduced a topographic constant, $A_{topo}$, relating the total landslide area to the area that excludes basins and inundated areas. We estimate $A_{topo}$ from the ALOS land cover. 97 % of all landslides occurred in areas without anthropogenic

influence, i.e. land with urban and agricultural use, and water bodies. We exclude water bodies, urban areas—predominantly the metropolitan area of Kumamoto City, and rice paddies from the topographic analysis, obtaining an affected area of 3037 km², i.e. $A_{topo} = 0.68$.





## 4   Total landslide area

Total landslide area is linked to several earthquake parameters, mostly magnitude and hypocenter or average rupture-plane depth. For instance, Keefer (1984) developed empirical relations, and Marc et al. (2017) introduced a model with physical assumptions. We adopted the relation by Marc et al. (2017) to check for completeness of the total landslide area of 6.38 km². The

actual landslide failure plane is likely smaller, as the NIED data set provides the combined area of depletion and accumulation. The modal hillslope inclination is estimated at 15°. Instead of the earthquake magnitude scaling relation (Leonard, 2010) used by (Marc et al., 2017), we use the rupture extent reported by Kubo et al. (2016). The area model requires the average length of the seismic asperities, which Marc et al. (2017) globally assumed as 3 km. However, Somerville et al. (1999) derived a relationship of asperity sizes based on seismic moment that results in an average asperity length of 12.78 km for the 2016

Kumamoto Earthquake. This length is consistent with the asperity sizes found by Yoshida et al. (2017) for their finite rupture model. The estimated landslide area with an asperity length of 3 km results in a predicted total landslide area of 12.90 km², while with the magnitude scaled asperity size of Somerville et al. (1999) the landslide area is 3.03 km². The landslide area estimates with constant asperity length and moment dependent asperity length differ by a factor of 2 and 0.5 from the NIED data set, respectively.

Landslide concentration is defined as landslide area per area at a given distance band (Meunier et al., 2007). For the seismic processing, we consider the rupture plane distance $r_{rup}$ based on the rupture model, instead of the hypocentral distance (Meunier et al., 2007) or the Joyner-Boore distance (Harp and Wilson, 1995).

## 5   Ground motion & seismically induced landsliding

### 5.1   Coseismic landslide displacement

The sliding-block model of Newmark (1965) is widely used to estimate coseismic hillslope performance (e.g. Kramer, 1996; Jibson, 1993, 2007). The model estimates the permanent displacement on a hillslope affected by ground motion. Newmark (1965) established a relation for hillslope displacement in terms of the maximum velocity at the hillslope for a single rectangular pulse, $v_{max}$ $\left[\mathrm{m\,s^{-1}}\right]$

$$d_s = \frac{v_{max}^2}{2}\left(\frac{1}{Aa_y} - \frac{1}{A}\right) \tag{3}$$

where $A$ is the magnitude of the acceleration pulse and $a_y$ $\left[\mathrm{m\,s^{-2}}\right]$ the yield acceleration, which is the minimum pseudostatic acceleration required to produce instability. For downslope motion along a sliding plane, $a_y$ is related to the angle of internal friction, $\phi_f$ and the hillslope inclination, $\delta$, by

$$a_y = g\left(\frac{\tan\phi}{\tan\delta}\right)\sin\delta = g(\overline{FS} - 1)\sin\delta \tag{4}$$

with the average factor of safety $\overline{FS}$. Chen et al. (2017) characterized unstable hillslopes by a safety factor of $FS < 1.5$.



An upper bound for the displacement $d_s$, is based on two ground motion parameters (Newmark, 1965; Kramer, 1996):

$$d_{max} = \frac{PGA}{a_y} \frac{PGV^2}{a_y},\tag{5}$$

where PGA $\left[\text{m s}^{-2}\right]$ and PGV $\left[\text{m s}^{-1}\right]$ are the peak ground acceleration and velocity, respectively.

## 5.2 Ground motion metrics

Though PGA is the most widely used ground-motion metric in geotechnical engineering, the Arias intensity $I_A$ (Arias, 1970) is widely used to characterize strong ground motion for landslides.

The Arias intensity is defined by

$$I_A = \frac{2}{\pi g} \int\limits_{T_1}^{T_2} a(t)^2 dt,\tag{6}$$

where $g = 9.80665$ m s$^{-2}$ is standard gravity and $T_1$ and $T_2$ are the times where strong ground motion starts and cedes. The
acceleration $a(t)$ is given in units of m s$^{-2}$ and the Arias intensity in m s$^{-1}$. The Arias intensity captures both the duration and amplitude of strong motion. Empirical relationships between $I_A$ and $d_s$ have been developed (e.g. Jibson, 1993; Bray and Travasarou, 2007; Jibson, 2007).

Since PGA and $I_A$ are related to each other (e.g. Romeo, 2000) and the hillslope displacement depends on both velocity and acceleration (Eq. (3), (5)), it is reasonable to characterize velocity similar to Arias intensity. The velocity counterpart to $I_A$ is
$IV2$, the integrated squared velocity (Kanamori et al., 1993; Festa et al., 2008):

$$IV2 = \int\limits_{T_1}^{T_2} v(t)^2 dt\tag{7}$$

The squared velocity is also used in radiated seismic energy estimates. The quantity $j_E$ is the radiated energy flux of an earthquake and estimated by (Choy and Cormier, 1986; Kanamori et al., 1993; Newman and Okal, 1998)

$$j_E = \frac{\rho c}{S_{amp}^2} e^{-kr_{rup}} IV2\tag{8}$$

where $\rho$ and $c$ are the density and seismic wave velocity at the recording site and $S_{amp}$ is the site specific amplification factor. The distance from the rupture is given by $r_{rup}$ and $k$ is a term for along-path attenuation (Anderson and Richards, 1975), and effects of transmission and reflection (Kanamori et al., 1993). The attenuation constant $k$ is also influenced by anisotropy and structure heterogeneity (Campillo and Plantet, 1991; Bora et al., 2015). The full definition of the energy flux includes two terms for compressional waves ($c = v_P$) and shear waves ($c = v_S$), respectively. The radiated energy of an earthquake, $E_S$, results
from the integral over the wavefront surface

$$E_S = \iint j_E dA,\tag{9}$$





where $A$ is the area of the surface through which the wave passes at the recording station and represents the geometrical spreading.

The radiated seismic energy $E_S$ describes the energy leaving the rupture area and is related to the seismic moment (Hanks and Kanamori, 1979)

$$E_S = \frac{\Delta\sigma}{2\mu}M_0, \tag{10}$$

where $\Delta\sigma$ is the stress drop and $\mu$ the shear modulus. We make use of this relation, when considering the magnitude related term in the ground motion model. Since most seismic energy is released as shear waves, we apply the shear wave velocity to the entire waveform and the site-specific correction term for the energy estimate, and the energy estimate, $\hat{E}$, based on Eq. (8) and (9) becomes

$$\hat{E} = \frac{\hat{A}\rho v_S}{S_{amp}^2}e^{-kr_{rup}}IV2 \tag{11}$$

While $E_S$ is the radiated seismic energy at the source, $\hat{E}$ is estimated from the velocity records at a station and only approximates $E_S$. Therefore, $\hat{E}$ may differ from the true and unknown radiated energy $E_S$ (Kanamori et al., 1993). Several assumptions characterize $\hat{E}$

– All energy is radiated as S-waves in an isotropic, homogeneous medium

– Geometrical spreading is corrected for an isotropic, homogeneous medium

– Since $IV2$ (Eq. (7)) depends on the radiation pattern, $\hat{E}$ depends on azimuth

– Attenuation is homogeneous

– Surface waves are not considered

– Site amplification is frequency-independent

Below, we investigate the azimuthal variation of the energy estimates to characterize the radiation pattern.

The estimated wavefront area $\hat{A}$ is related to the rupture extent and $r_{rup}$, and $\hat{A}$ corresponds to a simplified version of the wave front area approximation of Schnabel and Bolton Seed (1973); Shoja-taheri and Anderson (1988):

$$\hat{A} = 2WL + \pi r_{rup}(L + 2W) + 2\pi r_{rup}^2 \tag{12}$$

The extent of the rupture is assumed to be rectangular with length $L$ and width $W$. Equation (12) describes a cuboid with rounded corners with only half of its surface considered, because no energy flux is assumed to be transmitted above ground.

While the geometrical spreading correction is expressed analytically, in form of the wavefront area $\hat{A}$, we estimate the attenuation parameter $k$. Attenuation changes with distance as a power law at short distances ($< 150$ km) (Anderson and Richards, 1975) and longer distances are not considered. An empirical attenuation relationship is:

$$\ln Y = C + kr_{rup}, \tag{13}$$





where $Y$ is

$$Y = \frac{\hat{A}\rho v_S}{S^2_{amp}} \int\limits_{T_1}^{T_2} IV2, \tag{14}$$

i.e. the logarithm of the energy estimate without the attenuation term $e^{-kr_{rup}}$ from Eq. (11). The dummy variable $C$ is only used for estimating $k$ and not in the final correction for attenuation. A distance independent form of the Arias intensity, i.e.

corrected for geometrical spreading and attenuation, is defined by

$$I_{A,A} = \frac{\hat{A}}{S^2_{amp}} e^{-kr_{rup}} I_A, \tag{15}$$

where $k$ is determined by Eq. (13) and setting $Y = I_A \hat{A}$. The corrected Arias intensity $I_{A,A}$ is the acceleration based counter-part to $\hat{E}$.

Low-frequency effects, like directivity, are better captured with a velocity based metric (e.g. azimuth-dependent energy

estimate), than an acceleration based metric (Arias intensity) alone.

In terms of the Fourier transform, the sensitivity of acceleration at higher frequencies becomes apparent, as the Fourier transform of the time derivative of any function is

$$\mathcal{F}(\dot{f}(t)) = i\omega \mathcal{F}(f(t)) \tag{16}$$

and thus scales with frequency in the spectrum. The frequency sensitivity of $IV2$ and $I_A$ is related to the squared spectrum

given the squared nature of the metrics. As an example, we show in Fig. 3 the different spectral sensitivities of $IV2$ and $I_A$ for a theoretical seismic source spectrum (Brune, 1970). $IV2$, and thus $\hat{E}$ has a higher sensitivity to lower frequencies than $I_A$. The low-frequency part of the spectrum can be accounted for by considering $IV2$ in a ground-motion model.

### 5.3   Landslide related ground-motion models

The basic form of landslide related ground motion models for Arias intensity is based on earthquake magnitude $M$ and distance

$r$ (e.g. Harp and Wilson, 1995).

$$\ln I_A = c_1 + c_2 M + c_3 \ln r \tag{17}$$

This form is widely used (Keefer, 1984; Harp and Wilson, 1995). In engineering seismology, a typical ground motion model has an additional distance term for anelastic attenuation

$$\ln I_A = c_1 + c_2 M + c_3 r + (c_4 + c_5 M) \ln r \tag{18}$$

This is a modified version of the model template by Kramer (1996).

We exchange the magnitude term from Eq. (18) with a site-dependent energy term, as landsliding is more related to the energy of incoming seismic waves than to the moment at the source. We replace moment magnitude by the logarithm of energy



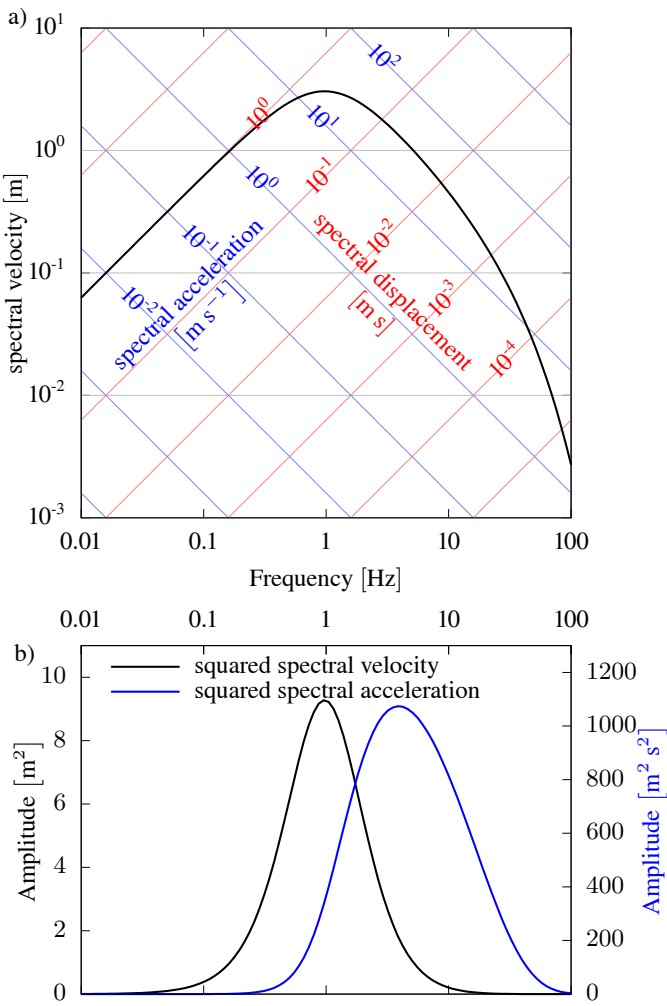

**Figure 3.** a) Far-field spectrum after Brune (1970). The spectrum can be read as displacement (red), velocity (black) and acceleration (blue). b) The squared Brune spectrum corresponds to the frequency sensitivity of velocity based $IV2$ (blue) and the acceleration based $I_A$ (black).





(Eq. (11)), since energy is proportional to the seismic moment $M_0$ (Eq. (9)). Based on the site-dependent energy estimate $\hat{E}$, we propose the model

$$\ln I_A = c_1 + c_2 \ln \hat{E} + c_3 r + (c_4 + c_5 \ln \hat{E}) \ln r \tag{19}$$

The five coefficients are inferred by non-linear least squares (e.g. Tarantola, 2005). We use the rupture plane distance ($r_{rup}$),

i.e. the shortest distance between a site and the rupture plane.

### 5.4   Rupture directivity model

In the NGA-west2 guidelines (Spudich et al., 2013), the directivity effect is modeled by isochrone theory (Bernard and Madariaga, 1984; Spudich and Chiou, 2008) or the azimuth between epicenter and site (Somerville et al., 1997). We use the latter approach and model directivity for estimated energy and corrected Arias intensity in a simplified way:

$$\ln \hat{E}_\theta = \ln \hat{E}_0 + a_E \cos(\theta - \theta_E) \tag{20}$$

$$\ln I_{A,A,\theta} = \ln I_{A,A,0} + a_I \cos(\theta - \theta_I) \tag{21}$$

$\hat{E}_0$ and $I_{A,A,0}$ are the offset (average), $a_E$ and $a_I$ the amplitude of variation with azimuth and $\theta_E$ and $\theta_E$ are the azimuths of the maximum. The definition of $\theta$ is similar to that of Somerville et al. (1997) as the angle measured between the epicenter and the recording site with the difference of being measured clockwise from north. The azimuths of the maximum, $\theta_E$ and $\theta_I$,

are free parameters for two reasons: (1) the rupture is assumed to have occurred on two faults and has thus variable strike, (2) the event is not pure strike-slip and has a normal faulting component. We therefore do not expect a match between the rupture strike and $\theta_E$ and $\theta_I$. The geometrical spreading is already incorporated in the energy estimate as a distance term (Somerville et al., 1997; Spudich et al., 2013).

### 5.5   Model for fault-normal to fault-parallel ratio

The ratio of the response spectra of the horizontal sensor components is a function of oscillatory frequency $f_{osc}$.

The north and east components ($E$, $N$) of the sensor are rotated to be fault-normal ($FN$) and fault-parallel ($FP$).

$$FN = E \cos \phi - N \sin \phi \tag{22}$$

$$FP = E \sin \phi + N \cos \phi \tag{23}$$

$$FN/FP = \frac{SA_{FN}(f_{osc})}{SA_{FP}(f_{osc})} \tag{24}$$

The response spectra are calculated from accelerograms after Weber (2002) with a damping of $\zeta = 0.05$.

The amplitudes of waves parallel to rupture propagation differ from waves normal to propagation on top of the directivity effect. This variation depends on the azimuth and is larger only at high periods like the directivity effect. The fault-normal response amplitude is larger than the fault-parallel response if directed (anti)parallel to the rupture. We model the ratio similar to Somerville et al. (1997)

$$\ln(FN/FP) = (b_1 + b_2 f_{osc}^{b_3} \cos(2(\theta - \theta_R))) H(b_1 + b_2 f_{osc}^{b_3}) \tag{25}$$




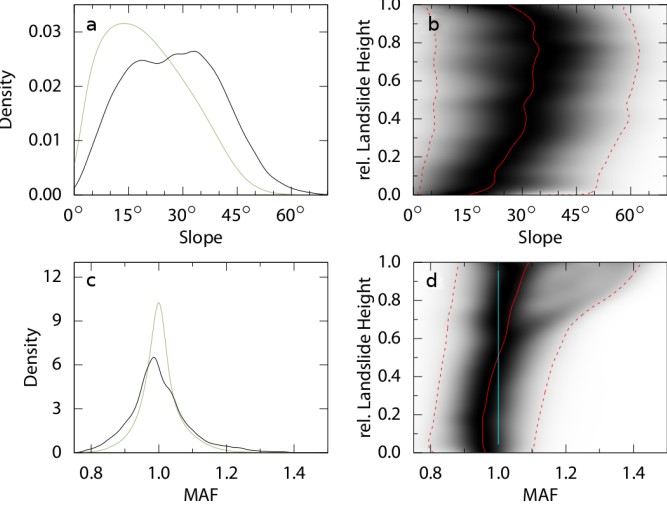

**Figure 4.** Distribution of hillslope inclination and MAF. a) Hillslope inclinations within the landslide affected area (gray), and within the landslide areas (black). b) Hillslope inclinations within all landslide areas with relative height: low height values (0.0 to ≈0.2) correspond to the area towards the landslide toe (tip is at 0.0). High values (≈0.8 to 1.0) represent the area close to the crown (top is 1.0). The solid red line is the mean; dashed red lines show the 5th and 95th percentiles. c) Same as a) but for MAF. d) Same as b) but for MAF. The cyan line shows MAF = 1, i.e. no amplification or attenuation.

where parameters $b_i$ describe the relationship of the oscillatory frequency to the ratio, $\theta$ is the azimuth (Eq. (20)), and $\theta_R$ is the azimuth of the maximal ratio. The ratio azimuth is subject to assumptions like its counterpart $\theta_E$. The Heaviside function $H(\cdot)$ avoids negative values in the model, which would be equivalent to an undesired phase shift in the cosine term.

We introduced a functional form for oscillatory frequency dependence with four parameters in Eq. (25). We did not introduce

5    a distance term and apply the model only to data with $r_{rup} \leq 50$ km.

## 6   Results

### 6.1   Topographic analysis

Landslides occurred mostly in tephra layers (Fig. 2a,b) covered by forests (Fig. 2d,e) and predominantly along the NE rupture segment. Nearly all landslides concentrated on hillslopes between 15° and 45° inclination and MAF ≈ 1 (Fig. 4a,c). Hillslope

10   inclination and MAF were higher towards the landslide crown (Fig. 4b,d), indicating a landslide failure process starting from the crown and according to simulations by Dang et al. (2016). They showed a progressive failure for two major landslides of Kumamoto earthquake by numerical simulation. TWI is linked to land cover and is highest in areas with rice paddies (Fig. 2i). Terrain with landslides has uniformly low TWI, thus we cannot evaluate the hydrological impact on the earthquake related landslides (e.g. Tang et al., 2018).





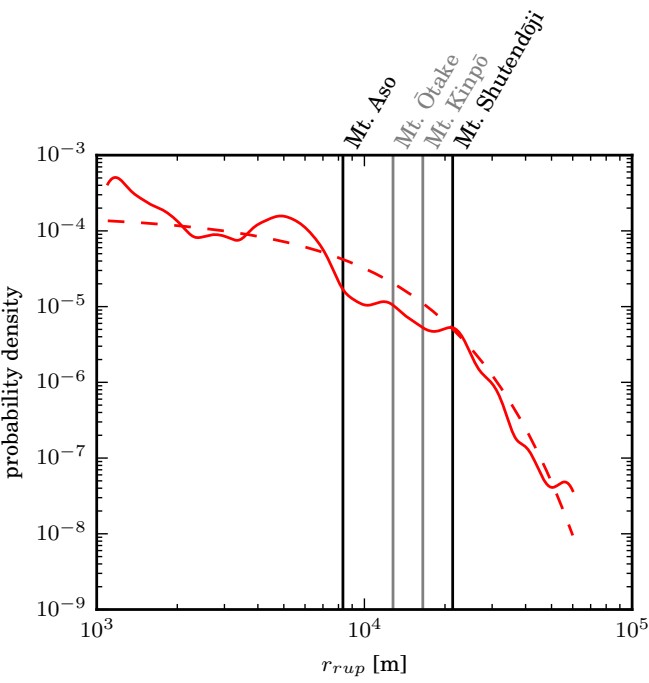

**Figure 5.** Kernel density estimate (solid red line) of landslide concentration with rupture distance $r_{rup}$ of the Kumamoto earthquake land-slides. The dashed red line is an exponential distribution with rate parameter estimated by maximum likelihood from landslide concentration.

Most landslides originated at localities with amplified ground accelerations and steep hillslopes and propagated progressively to flatter areas with less amplified ground accelerations. Landslides—interpreted as shear failure—start as mode II (in-plane shear) failure at the scarp and mode III (anti-plane shear) failure at the flanks (McClung, 1981; Fleming and Johnson, 1989; Martel, 2004). At later stages of the landslide rupture mode I (widening) failure can co-occur the process (Martel, 2004).

5 Simulations of elliptic landslides by Martel (2004) show that the most compressive and the most tensile stresses are parallel to the major axis of the landslide, coinciding with the average landslide aspect. Yamada et al. (2013) and Yamada et al. (2018) show for several japanese landslides that peak forces were aligned parallel to the long side of the landslides; Allstadt (2013) shows from waveform inversion for the Mt. Meager landslide that force and acceleration were parallel to the longer side of the landslide source area.

10 Mt. Aso and its caldera and Mt. Shutendōji had a high density of landslides (Fig. 5), whereas Mt. Kinpō and Mt. Ōtake lack landslides, though these locations are closer to the epicenter and at comparable distances from the rupture (Fig. 5). However, all these locations have the same rock type and land cover, comparable hillslope inclination and MAF. Hence, lithology, land cover, and topographic characteristics are insufficient to explain the landslide distribution and concentration with respect to the hypocenter, because both differ widely despite near-identical conditions.



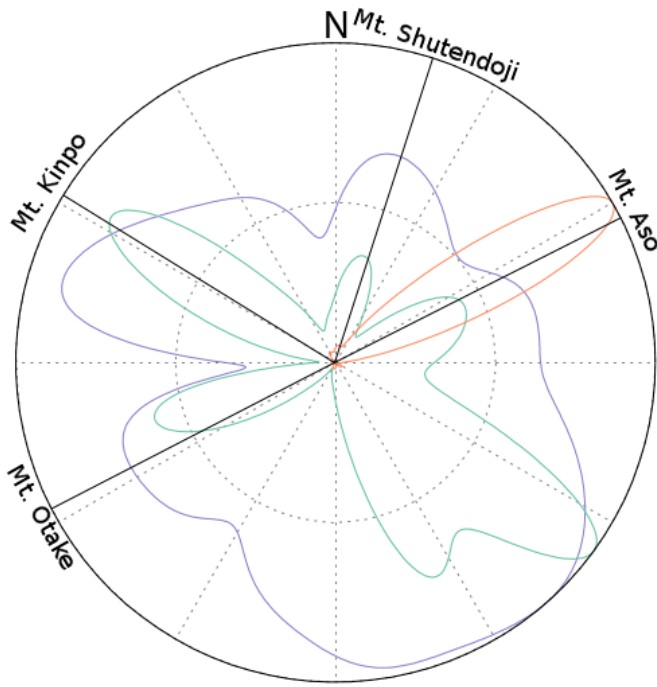

**Figure 6.** Density of hillslope inclinations with azimuth of the Kumamoto earthquake landslides (orange), unspecified landslides within the landslide affected area (green), and the entire landslide affected area (blue). The azimuth is measured with respect to the earthquake epicenter (see Fig. 7) and densitites are normalized to their maxima.

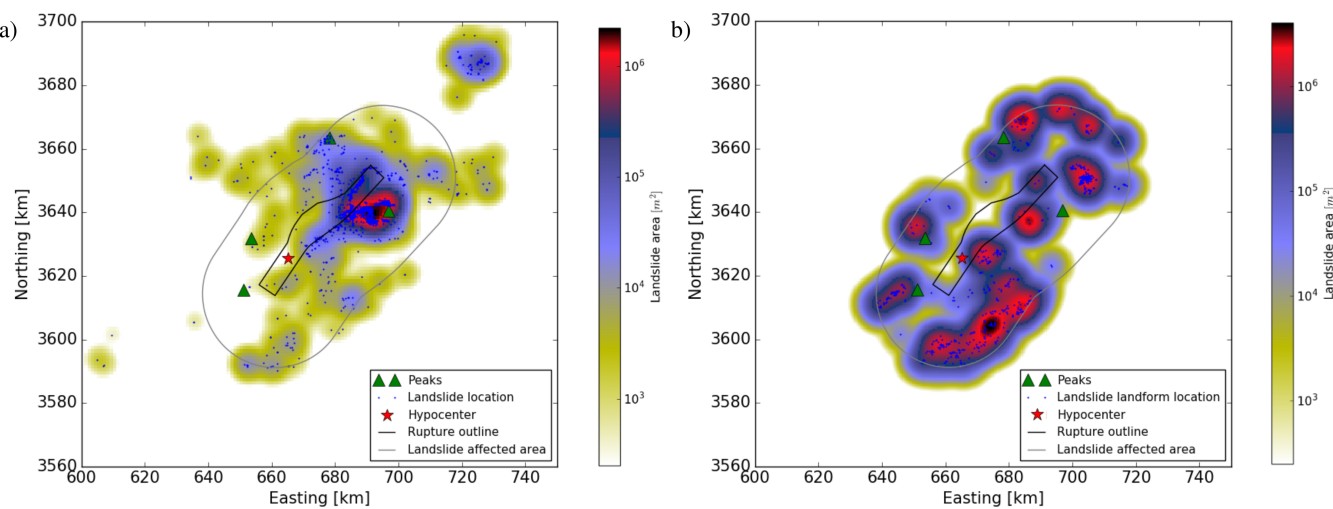

**Figure 7.** Spatial distribution of landslides. a) Coseismic landslides. The total landslide area at a location is shown as a color-coded smooth function in the background. b) Same as in (a) but for unspecified landslides within the landslide affected area of the Kumamoto earthquake.



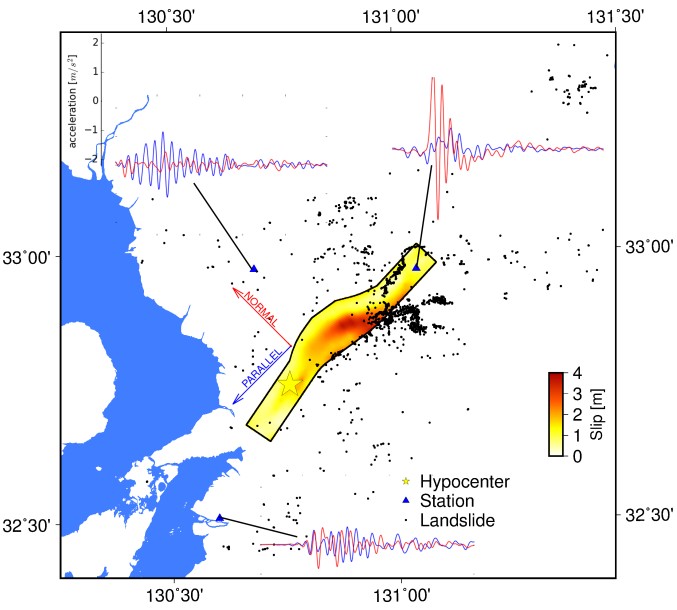

**Figure 8.** Characteristic waveforms observed in the vicinity of the rupture. The waveforms shown are low-passed filter at 1.2 Hz.

The azimuthal density—with respect to the epicenter—of the unspecified landslides follows to some extent the distribution of hillslope inclinations $\geq 19°$ in the landslide affected area (Fig. 6). This similarity shows that the abundance of unspecified landslides mimics the steepness of topography in the region. Densities are higher towards Mt. Kinpō (NW), Mt. Ōtake (WSW), Mt. Shutendōji (N), Mt. Aso (NE), and the Kyūshū Mts. (SE). The coseismic landslide distribution differs completely from the distributions of unspecified landslides and topography, respectively, as nearly all landslides occurred to the northeast of the epicenter close to the rupture plane (Fig. 7). The contrast between the distributions of unspecified landslides and earthquake related landslides indicates a contribution by the rupture process.

## 6.2 Impact of finite source on ground motion and landslides

The results of the seismic analysis are given for waveforms, the basis for $\hat{E}$ and $I_A$, and response spectra, used for $FN/FP$. To the northeast, signals with forward-directivity are shorter in duration with one or few strong pulses (Fig. 8, top right). Waveforms with backward-directivity to the southwest of the rupture are longer with no dominant pulse (Fig. 8, bottom left). Waveforms parallel to the rupture have intermediate duration. Waveforms in either forward- or backward-direction have stronger amplitudes in the fault-normal direction, whereas waveforms outside the directivity-affected regions have stronger amplitudes in the fault-parallel direction (Fig. 8, top left).

We estimated energies $\hat{E}$ from the three-component waveforms. For the Arias intensity, both horizontal components are used. The geometrical spreading $A$ is calculated according to Eq. (12) with a rupture length of $L = 53.5$ km and width of $W = 24.0$





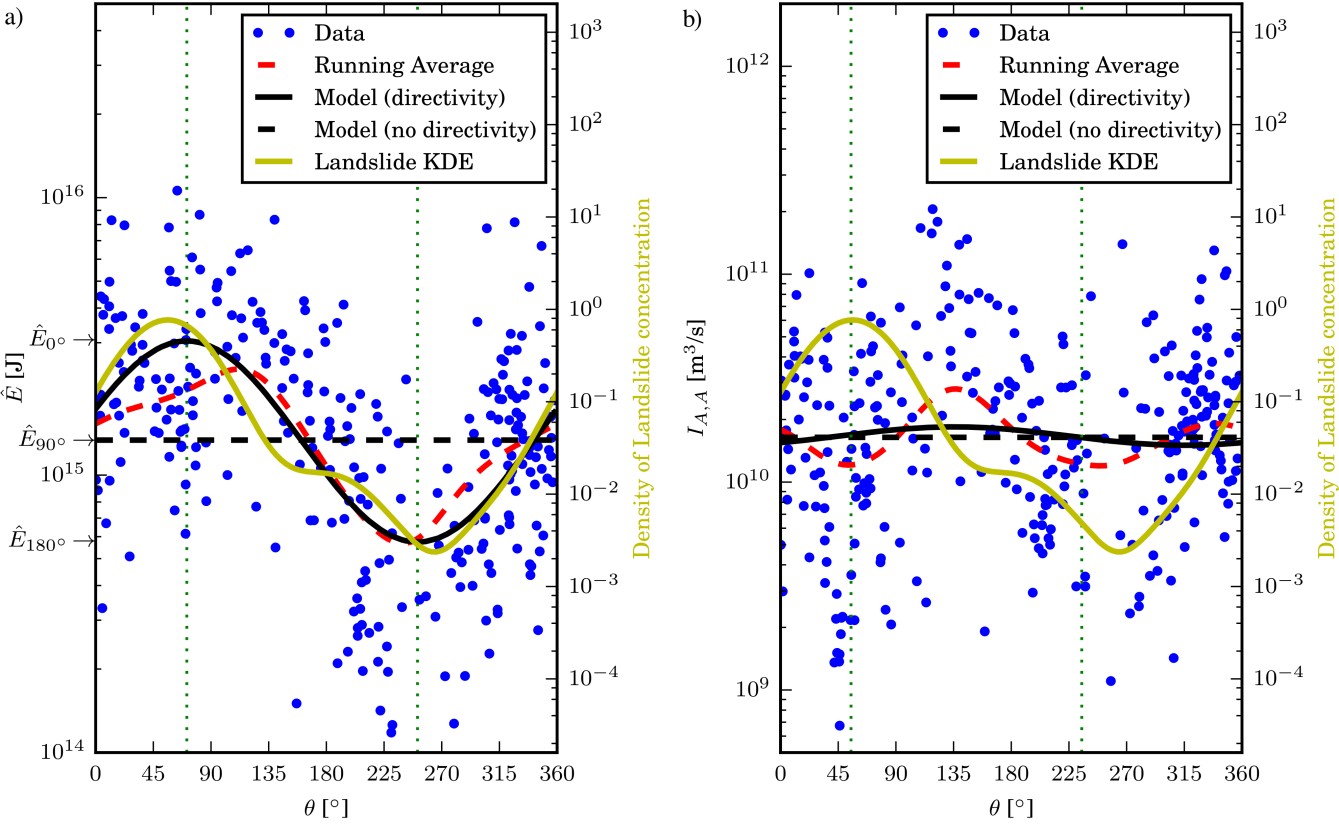

**Figure 9.** a) Energy estimates ($\hat{E}$) over azimuth. b) Same as in a) but for Arias intensity with correction for geometrical spreading ($I_{A,A}$).

km. Any remaining distance dependence has been corrected for by estimating and applying the attenuation parameter $k$ (Eq. (13))

After the determination of $k$, $\hat{E}$ and $I_{A,A}$ are considered distance independent and can be investigated for azimuthal variations. With a reference point for the azimuth at the epicenter, $\hat{E}$ shows oscillating variations in amplitude with azimuth (Fig.

5 9a), while $I_{A,A}$ exhibits a similar amplitude variations over the entire azimuthal range (Fig.9b). The running average based on a von Mises kernel ($\kappa_{vM} = 50$) of $\hat{E}$ and $I_{A,A}$ shows increased $\hat{E}$ between 45° and 135°, i.e. approximately parallel to the strike. Minimal values of $\hat{E}$ occur in the opposite direction (200° - 300°). The running average of $I_{A,A}$ shows several fluctuations but not as wide and large as that of $\hat{E}$. The azimuthal variation of $\hat{E}$ indicates the rupture directivity and the absence of large variations in $I_{A,A}$ shows that the directivity effect is only evident at lower frequencies (compare with Fig. 3).

10 The azimuthal variation of $\hat{E}$ and $I_{A,A}$ is modelled according to Eq. (20). We estimate parameters for two scenarios:

- directivity is assumed, resulting in azimuthal variations and $a_E$ and $a_I$ are free parameters,

- directivity is not assumed, resulting in no azimuthal variations with $a_E = a_I = 0$.





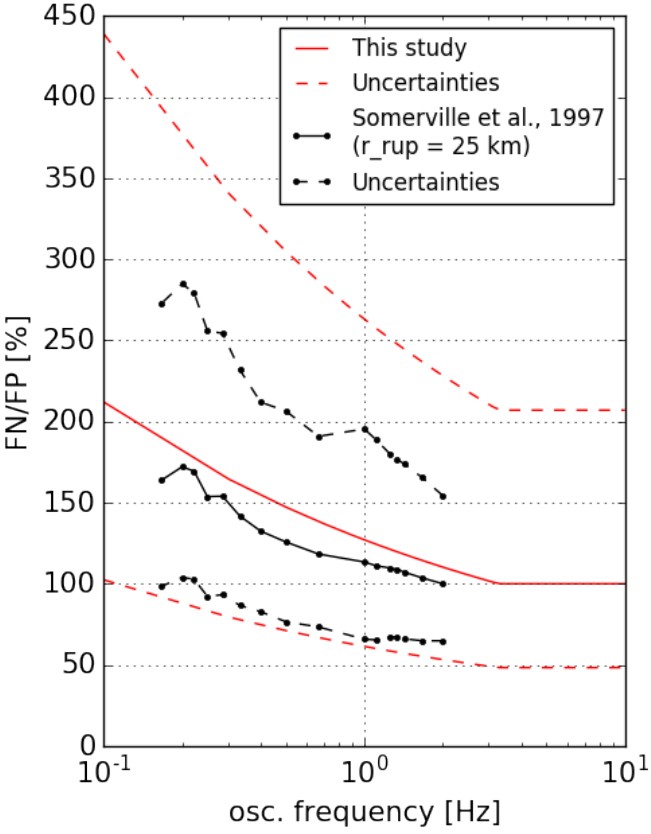

**Figure 10.** Model for the amplitude ratio of response spectra of fault-normal and fault-parallel components ($FN/FP$) as function of oscillatory frequency.

The two models are compared with the Bayesian Information Criterion (BIC, Schwarz, 1978) for a least squares fit:

$$BIC = n \ln N + N \ln \hat{\sigma}^2, \tag{26}$$

where $n$ is the number of estimated parameters ($n = 4$ for first case, $n = 2$ for second case), $N$ is the number of data, and $\hat{\sigma}^2$ is the variance of the model residuals. The model with the smaller BIC is preferred. The starting values of the parameters are the

5   mean of $\hat{E}$ and $I_{A,A}$, no azimuthal variation ($a_d = 0$), and the azimuths of the maximum of $\hat{E}_\theta$ and $I_{A,A,\theta}$ are set to the strike of the fault ($\theta_E = \theta_I = 225°$).

   The directivity model for $\hat{E}$ follows the trend of the data and the running average closer than the model without directivity (Fig. 9a). According to BIC, the model with directivity is preferable ($BIC_{\text{directivity}} = -110$, $BIC_{\text{no directivity}} = -11$). In case of the Arias intensity, the difference in BIC between the two models is less compared to the azimuth-dependent energy (Fig. 9b).

10   Here, the model without directivity is the preferred one ($BIC_{\text{directivity}} = 30$, $BIC_{\text{no directivity}} = 22$). In consequence, azimuthal variations in wave amplitudes and energy related to the directivity effect occur at lower frequencies.





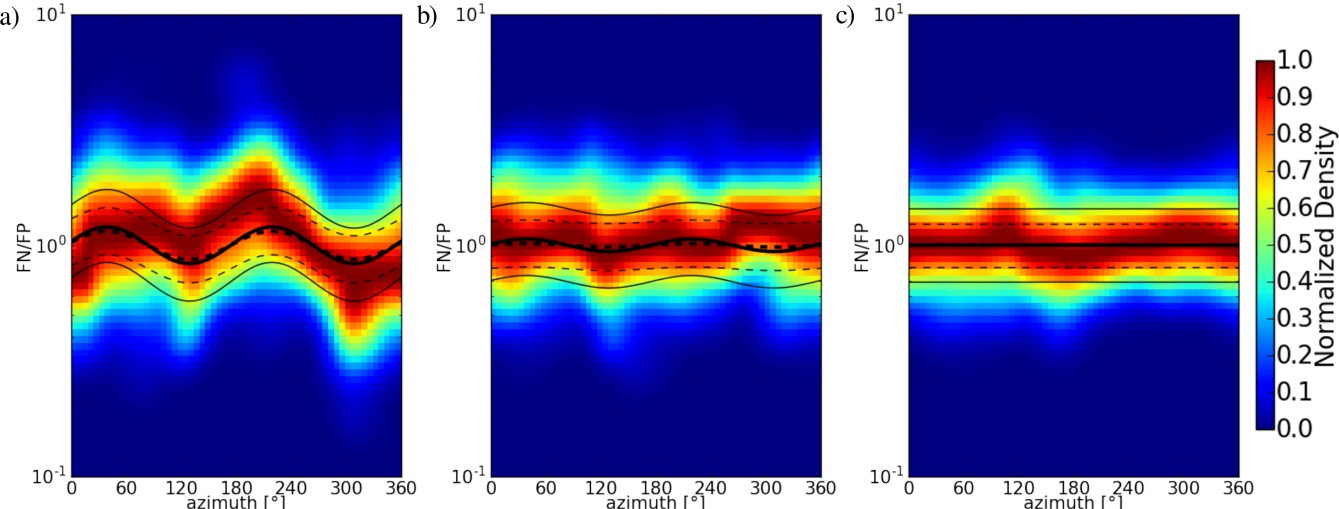

**Figure 11.** Density of $FN/FP$ with azimuth obtained from response spectra for three different oscillatory frequency ranges: a) 0.1 - 1 Hz, b) 1.0 - 2.5 Hz, c) > 2.5 Hz. For each plot, our FN/FP model and the model of Somerville et al. (1997) are shown for a) 0.55 Hz, b) 1.75 Hz, c) 4 Hz.

The forward directivity waves contain a very strong low frequency pulse (Fig. 8). The pulse amplitude depends on the ratio of rupture and shear wave velocity and the length of the rupture (Spudich and Chiou, 2008). The forward directivity pulse is superimposed by high frequency signals in acceleration traces but becomes more prominent in velocity traces (Baker, 2007), due to its low frequency nature, i.e. below 1.6 Hz (Somerville et al., 1997).

5     The low-frequency azimuthal variations are also reflected in the spectral response of the waveforms. Spectral accelerations of stations with $r_{rup} \leq 50$ km were computed from 0.1 Hz - 5 Hz with an interval of 0.01 Hz for the fault-normal and fault-parallel component. The distribution of $FN/FP$ shows decreasing azimuthal variability with increasing oscillatory frequency (Fig. 11a). $FN/FP$ shows highest variation with azimuth at low oscillatory frequencies (0.1 - 1 Hz, Fig. 11a); variations are much smaller between 1 and 2.5 Hz (Fig. 11b); and nearly absent above 2.5 Hz (Fig. 11c). This decrease with frequency is captured
10     by the $FN/FP$ model (Eq. (25), Fig. 10). Since our model is an average over the covered distance, with an average rupture distance of 25.06 km, we compare it to the $FN/FP$ model of Somerville et al. (1997) at 25 km (Fig. 11). Both models show a similar decay with frequency with our model predicting a slightly higher $FN/FP$. Therefore, the wave polarity ratio related to rupture directivity is pronounced at lower frequencies and dissipates with increasing frequency, similar to the azimuthal variations observable in energy estimates (lower frequencies) but not in Arias intensity (higher frequencies).

15     The pattern of low-frequency ground motion is well reflected in that of landslides. The azimuthal variation of $\hat{E}$ coincides with that of landslide concentration (Fig. 9). Both azimuth-dependent energy and landslide concentration have a similar trend with the maximum parallel to rupture direction and the minimum strike anti-parallel. The orientation of maximum $FN/FP$ is also reflected in the landslide aspect. Two directions show higher landslide density, one to the northwest and one to the east (Fig. 12a). The highest density of landslides has a northwestern aspect in agreement with maximum $FN/FP$, both perpendicular





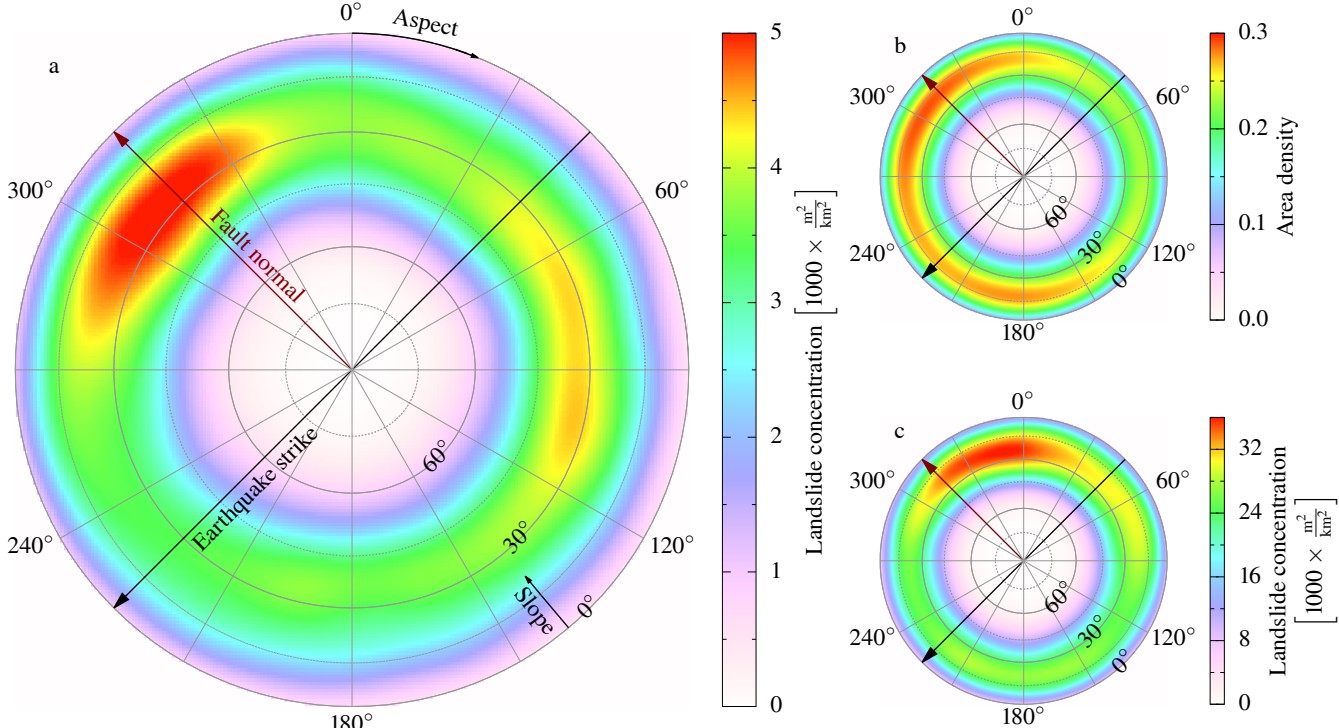

**Figure 12.** a) Aspect and hillslope inclination distribution of the hillslopes within the areas of the earthquake triggered landslides. This distribution is normalized by the distribution of the aspect in the landslide affected area b). The black line denotes the strike of the Kumamoto earthquake (225°) b) Distribution of aspect and hillslope inclination in the landslide affected area. c) Same as in a) but for unspecified landslides.

to the strike. The eastward increased density is mostly due to landslides very close to the rupture. A look at different distances reveals that the increased density of landslides with aspect east by southeast occurs at very short distances ($r_{rup} \leq 2.5$ km, Fig. 14), while the northwest oriented landslides are further away (2.5 km $< r_{rup} \leq 6$ km). Only minor landslides have larger distances with no specific pattern.

5    The distribution of aspect and hillslope inclination in the landslide affected area shows little variation over the entire aspect range (Fig. 12b) with slightly more hillslopes facing westward. Neither the highly preferred orientation of landslides to the northwest nor to the east is reflected by the topography of the landslide affected area (12a,b). The unspecified landslides within the affected area have a near northward aspect and deviate by $\approx 30°$ from the earthquake triggered landslides (Fig. 12c). This highlights that the earthquake not only affects the landslide locations (Fig. 6, 7), but also their orientation (Fig. 12).

## 10  6.3   Ground motion model for Kumamoto

We derived two ground-motion models for Arias intensity from data with $r_{rup} \leq 150$ km (Tab. 1, Fig. 15). One model incorporates the azimuth-dependent seismic energy (Eq. (19)). The other model is a conventional isotropic moment magnitude



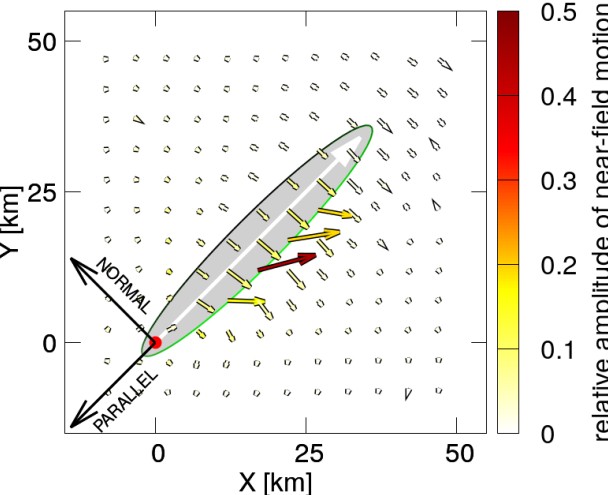

**Figure 13.** Orientation of horizontal peak-ground acceleration for the simulated waveforms. The arrow length scales with magnitude of acceleration. The simulated rupture plane is oriented as the rupture plane of the Kumamoto earthquake (strike: $225°$, dip: $70°$) and of elliptic shape (gray). The upper side is denoted by the green line, the lower half by black. The rupture process originated at the hypocenter (red dot) with circular propagation outwards (white arrow).

| | model using $\hat{E}$ | model using $M_W$ |
|---|---|---|
| $c_1$ | 4.083 | 5.879 |
| $c_2$ | $1.162 \times 10^{-1}$ | $-3.201 \times 10^{-2}$ |
| $c_3$ | $-3.052 \times 10^{-5}$ | $-3.172 \times 10^{-5}$ |
| $c_4$ | $-4.343 \times 10^{-1}$ | $-2.349 \times 10^{0}$ |
| $c_5$ | | $5.565 \times 10^{-2}$ |

**Table 1.** Parameters for ground-motion models





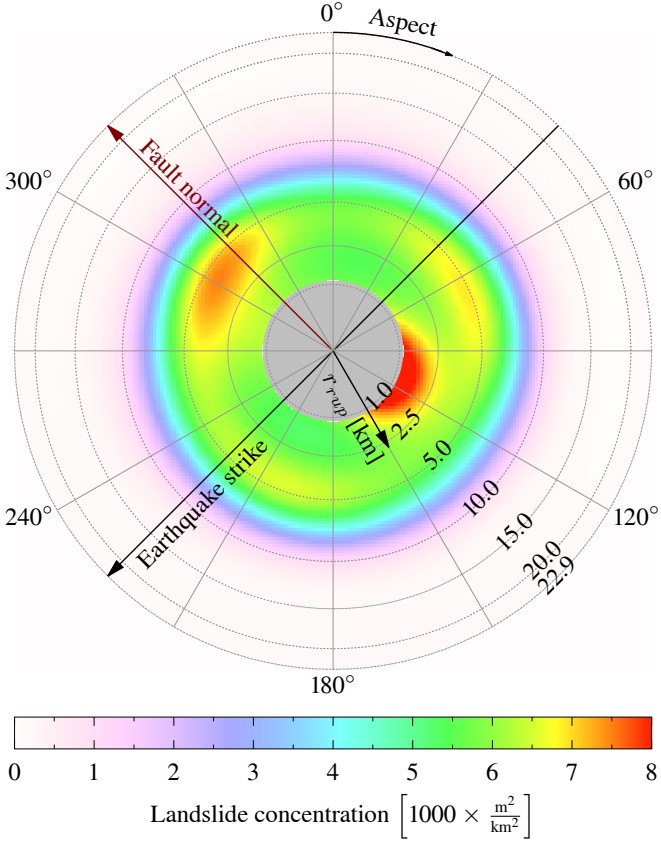

**Figure 14.** Distribution of landslides with aspect and rupture distance. The rupture distance is measured from the model of Kubo et al. (2016). This model does not completely reach the surface, hence no distance below 1 km are present. The distribution has been normalized by the distribution of aspect of the affected area.

dependent model (Eq. (18)). The decay of Arias intensity with distance for both models fits well the running average and is proportional to the decrease in landslide density with distance. Variation of estimated energy is well covered by the model and spans more than two orders of magnitude resulting in variation of Arias intensity of nearly one order of magnitude.

The magnitude based model is nearly equivalent to the energy based model with $\hat{E} = 1.2 \times 10^{15}$ J. This value is close to the

5    average energy estimate found from energy estimates of the directivity model from Eq. (20) ($\hat{E} = 1.3 \times 10^{15}$ J). The closeness of the two values implies that the magnitude based model can be seen as an average over the azimuth of the energy based model.

## 7    Discussion

We provide a framework for characterizing coseismic landslides with an integrated approach of geomorphology and seismology

10   emphasizing here the role of low-frequency seismic directivity and finite source. Given the observations of ground motion of





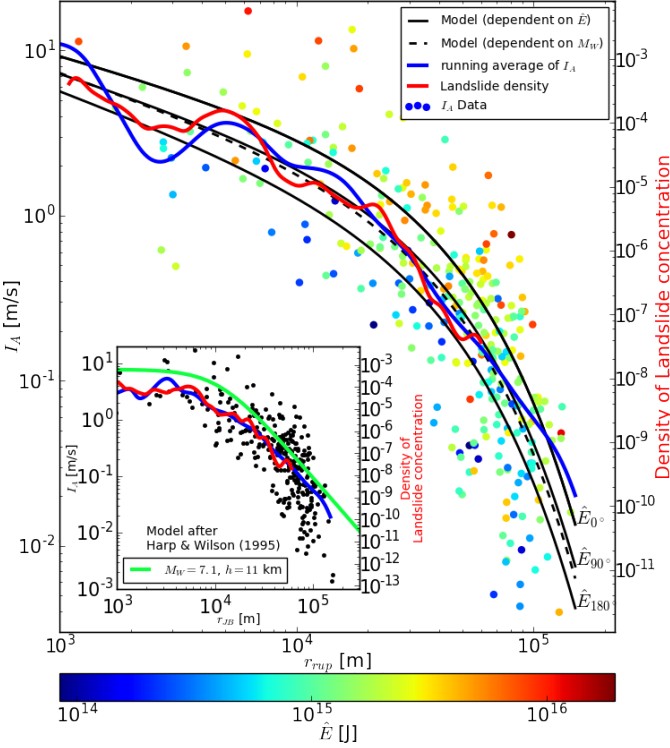

**Figure 15.** Ground motion model for $I_A$. The solid lines represent the model with energy estimates for three different energy levels as in Fig. 9a. The inset figure shows for comparison the ground motion model of Harp and Wilson (1995) (green), landslide concentration density (red).

the Kumamoto earthquake, two questions arise: (1) How specific is the observed ground motion, i.e. is the Kumamoto rupture particularly distinct? (2) As a rupture very close to the surface, how much does seismic near-field motion contribute? The second question arises, because many landslides occurred very close to the rupture plane. However, it is not possible to separate the observed waveforms into near-, intermediate-, and far-field terms. To investigate both questions, we computed theoretical

5  waveforms after Haskell (1964); Savage (1966); Aki and Richards (2002) from a circular rupture on an elliptic finite source with constant rupture velocity in a homogeneous, isotropic, and unbound medium (see Appendix).

Despite the simplified assumptions behind this waveform model, low-frequency ground motions capture the most prominent features of the observed waveforms. Simulated waveforms close to the rupture plane show a change in polarity orientation towards east-west, while the strong fault-normal polarity appears at larger distances. A decomposition into a near-field term

10  and combined intermediate- and far-field term reveals that the former highly contributes to the ground-motion at short distances. The impact of the near-field term may explain the increase in eastward aspect (Fig. 14) of landslides close to the rupture.

The simulations also demonstrate the effect of directivity on estimates of radiated energy and Arias intensity. The azimuthal variations of simulated $\hat{E}$ are similar to the observed variations. The Arias intensity of the simulations also displays azimuthal



variations with same characteristics as the energy estimate. These variations in Arias intensity are absent in the observed data, indicating that Arias intensity is more influenced by local heterogeneities and scattering than the energy estimates as these are ignored in the simulations.

The results show that the Arias intensity is not as susceptible to the directivity effect and variations in fault-normal to fault-parallel amplitudes as is the radiated energy: Because of its higher sensitivity towards higher frequencies, these effects are masked by high-frequency effects, e.g. wave scattering and a heterogeneous medium. We found that the radiation pattern related to the directivity effect is recoverable from energy estimates but not from Arias intensity. This low-frequency dependence has also been demonstrated by the response spectra ratios for $FN/FP$ where directivity related amplitude variations with azimuth have been identified only for frequencies $< 2$ Hz which is in agreement with previous work (Spudich et al., 2004; Somerville et al., 1997). We introduced a modified model for Arias intensity using site-dependent seismic energy estimates instead of the source-dependent seismic magnitude to better capture the effects of low-frequency ground motion.

The conventional magnitude based isotropic model and the azimuth-dependent seismic energy model correlate with the landslide concentration over distance (Fig. 15). As in Meunier et al. (2007) it is therefore feasible to use the ground-motion model to model the landslide concentration, $P_{ls}(r_{rup})$, by a linear relationship

$$\ln P_{ls}(r_{rup}) = a_r + b_r \ln r_{rup} \tag{27}$$

Azimuthal variations of landslide density, $P_{ls}(\theta)$, correspond to azimuthal variations in seismic energy and can be described by a similar relationship

$$\ln P_{ls}(\theta) = a_\theta + b_\theta \cos(\theta - \theta_0) \tag{28}$$

For the Kumamoto earthquake data we estimate $a_r = -5.4$, $b_r = 2.6$ and $a_\theta = -46.8$, $b_\theta = 3.0$. The azimuth-dependent landslide concentration implies that similar landslide concentrations can occur at different distances from the rupture, thus partly explaining some of the deviation in Fig. 5 and Fig. 15.

Compared to the model of Harp and Wilson (1995) (Fig. 15) our model uses rupture-plane distance, as opposed to the Joyner-Boore distance ($r_{JB}$). When using the hypocentral depth as pseudodepth, the model of Harp and Wilson (1995) overpredicts $I_A$ both at shorter and longer distances—irrespective of the pseudodepth at larger distances. This is most likely due to the lack of an additional distant dependent attenuation term in their model (Eq. (17)).

The use of MAF instead of curvature alone provides a proxy by how much a seismic wave is amplified (or attenuated) for a given wavelength and location. We showed that both hillslope inclination and MAF tend to be lower towards the landslide toe (Fig. 4). This effect is linked to the convention that landslide polygons cover both the zone of depletion and accumulation. When relating coseismic landsliding to the seismic rupture, only the failure plane of the landslide matters, because this is the hillslope portion that failed under seismic acceleration. For instance, Chen et al. (2017) found that landslide susceptibility and safety factor calculation depends on whether the entire landslide or only parts—scarp area or area of dislocated mass—are considered. It is intractable to reconstruct each individual landslide failure plane from the mapped data. However, failure may have likely originated close to the crown and then progressively propagated downward the hillslope, because MAF $> 1$





indicates an amplification of ground motion towards the crown of the landslides. Additionally, (Sato et al., 2017) consider the tephra layers rich in halloysite to be the main sliding surfaces indicating shallow landslides (Song et al., 2017).

Coseismic landslide locations have a uniformly low topographic wetness index, indicating that hydrology may have added little variability to the pattern of the earthquake triggered landslides, i.e. we could not trace any clear impact of soil moisture.
Though soil moisture might influence the coseismic landslide pattern (Tang et al., 2018).

## 8 Conclusions

We investigated the Kumamoto earthquake and its triggered landslides. We demonstrated that the pattern of coseismic land-slides is consistent with that of low-frequency ground motion. At low frequencies it exhibits two aspects: rupture directivity and increased amplitudes normal to the fault. While the former effect influences landslide locations, the latter effect relates
to the landslide orientation (aspect). Topographic controls (hillslope inclination and MAF) are limited predictors of coseismic landslide occurrence, because areas with similar topographic and geological properties at similar distances from the rupture showed widely differing landslide activity (Havenith et al., 2016; Massey et al., 2018). Nonetheless landslides concentrated only to the northeast of the earthquake rupture, while unspecified landslides have been identified throughout the affected region.

We introduced a modified model for Arias intensity using site-dependent radiated seismic energy estimates instead of the
source-dependent seismic magnitude to better model low-frequency ground-motion.

Compared to previous models widely used in landslide related ground motion characterization our model is based on state-of-the-art ground-motion models used in engineering seismology, which have two different distance terms, one for geometrical spreading and one for along-path attenuation. The latter is not commonly incorporated in landslide studies (e.g. Meunier et al., 2007; Massey et al., 2018). Our results emphasize that the attenuation term should be considered in ground-motion models, as
the landslide concentration with distance shows similarities to such ground-motion models.

The effect of low-frequency ground motion on the rupture process of the landslides results in landslide movements parallel to highest ground-motion. Due to the surface proximity of the earthquake rupture plane, near-field ground motion influences the aspect of close landslides to be east-southeast. The intermediate- and far-field motion of the earthquake promoted more landslides on northwest exposed hillslopes, an effect that overrides those of steepness and orientation of hillslopes in the region.
We demonstrated that earthquake triggered landslide hazard estimation requires an integrated approach of both detailed ground-motion and topographic characterization. While the latter is well established for landslide hazard, ground-motion char-acterization has been only incorporated by simple means, i.e. not incorporating azimuth-dependent finite rupture effects. Our results for the Kumamoto earthquake demonstrate that seismic waveforms can be reproduced by established methods from seismology. We suggest that these methods can improve landslide hazard assessment by including models for finite rupture
effects.



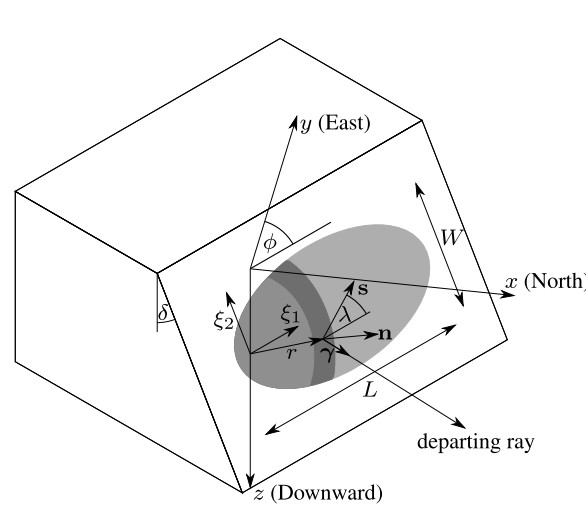

**Figure A1.** Setup of the rupture model. Gray ellipse represents the rupture: light gray area is unruptured, medium gray area is slipping, and the dark gray area is after slip arrest.

## Appendix A: Synthetic waveforms from displacement of a finite rupture

We illustrate the generation of ground displacement as a discontinuity across a rupture fault (e.g. Haskell, 1964, 1969; Anderson and Richards, 1975; Aki and Richards, 2002). The displacement for any point $x$ at time $t$ is given by

$$u_i(\mathbf{x},t) = \iint\limits_{\Sigma} c_{jkpq} \frac{\partial G_{ip}(D_j(\boldsymbol{\xi},t))}{\partial x_q} n_k d\Sigma \tag{A1}$$

5    where $c$ is the fourth order elasticity tensor from Hooke's law, $G$ is the Green's function describing the response of the medium, $\mathbf{D}(\boldsymbol{\xi},t)$ is the displacement on the fault with area $\Sigma$ and coordinates $\boldsymbol{\xi}$, $\mathbf{n}$ is the fault normal vector. Summation over $i,j,p,q$ is implied. While the surface integral is carried out numerically, the derivatives of the Green's function for an isotropic, homoge-





neous, and unbound medium can be solved analytically.

$$\frac{\partial}{\partial x_q} G_{ip}(D_j(\boldsymbol{\xi},t)) = \tag{A2a}$$

$$\frac{15\gamma_i\gamma_p\gamma_q - 3(\delta_{ip}\gamma_q + \delta_{iq}\gamma_p + \delta_{pq}\gamma_i)}{4\pi\rho r^4} \int_{\frac{r}{\alpha}}^{\frac{r}{\beta}} D_j(\boldsymbol{\xi},t-\tau)\tau d\tau \tag{A2b}$$

$$+\frac{6\gamma_i\gamma_p\gamma_q - (\delta_{ip}\gamma_q + \delta_{iq}\gamma_p + \delta_{pq}\gamma_i)}{4\pi\rho\alpha^2 r^2} D_j\left(\boldsymbol{\xi},t-\frac{r}{\alpha}\right) \tag{A2c}$$

$$-\frac{6\gamma_i\gamma_p\gamma_q - (2\delta_{ip}\gamma_q + \delta_{iq}\gamma_p + \delta_{pq}\gamma_i)}{4\pi\rho\beta^2 r^2} D_j\left(\boldsymbol{\xi},t-\frac{r}{\beta}\right) \tag{A2d}$$

$$+\frac{\gamma_i\gamma_p\gamma_q}{4\pi\rho\alpha^3 r} \dot{D}_j\left(\boldsymbol{\xi},t-\frac{r}{\alpha}\right) \tag{A2e}$$

$$-\frac{\gamma_i\gamma_p\gamma_q - \delta_{ip}\gamma_q}{4\pi\rho\beta^3 r} \dot{D}_j\left(\boldsymbol{\xi},t-\frac{r}{\beta}\right) \tag{A2f}$$

where

$$r = |\mathbf{x} - \boldsymbol{\xi}| \text{ and } \gamma_i = \frac{x_i - \xi_i}{r} \tag{A3}$$

and $\delta_{ij}$ is Kronecker's delta. The terms in Eq. (A2) are commonly separated in groups with respect to their distance $r$. In Eq. (A2a) is the near-field (NF) term, as its amplitude decays with $r^{-4}$, it affects the immediate vicinity of a rupture only. Terms with a distance attenuation proportional to $r^{-2}$ are called intermediate-field (IF) terms for $P$-waves (Eq. (A2c)) and $S$-waves (Eq. (A2d)). The remaining two terms are the far-field (FF) terms for $P$-waves (Eq. (A2e)) and $S$-waves (Eq. (A2f)) with a decay proportional to $r^{-1}$. A major difference between the NF and IF terms, and the FF terms is that the former depend on the slip on the rupture and they are the cause for static and dynamic displacement; whereas the latter are functions of the time derivative of slip and result in dynamic displacement only.

The slip function in time is related to the Yoffe function Yoffe (1951); Tinti et al. (2005) with rise time $T$. We use the slip distribution of Savage (1966) to describe the amplitude distribution of the slip on the rupture, as well as the elliptical fault shape and rupture propagation from Savage (1966). The slip amplitude is given by

$$D(\boldsymbol{\xi}) = D_0\sqrt{1 - \left(\frac{\xi_1 - p\epsilon\frac{L}{2}}{\frac{L}{2}}\right)^2 - \left(\frac{\xi_2}{\frac{W}{2}}\right)^2} \tag{A4}$$

where $D_0$ is the maximum displacement at the center of the fault, $L$ and $W$ are the length and width of the fault, and $p$ determines whether the rupture starts at the focus at the front of the rupture plane (strike-parallel, $p = 1$) or at the focus at the end (strike-anti-parallel, $p = -1$). The rupture originates in one of the two foci and propagates radially away from the source with constant velocity $\zeta$ and terminates when it reaches the rupture boundary. The slip vector $\hat{\mathbf{s}}$ describes the orientation of the displacement $D(\boldsymbol{\xi})$ on the fault plane. We follow the definition of $\hat{\mathbf{n}}$ and $\hat{\mathbf{s}}$ in terms of fault strike $\phi_s$, dip $\delta$, and rake $\lambda$ from



Aki and Richards (2002):

$$\hat{\mathbf{n}} = \begin{pmatrix} -\sin\delta\sin\phi_s \\ \sin\delta\cos\phi_s \\ -\cos\delta \end{pmatrix} \tag{A5}$$

$$\hat{\mathbf{s}} = \begin{pmatrix} \cos\lambda\cos\phi_s + \cos\delta\sin\lambda\sin\phi_s \\ \cos\lambda\sin\phi_s - \cos\delta\sin\lambda\cos\phi_s \\ -\sin\lambda\sin\delta \end{pmatrix} \tag{A6}$$

The displacement vector $\mathbf{D}$ in Eq. (A2) is given by

$$\mathbf{D} = D(\boldsymbol{\xi})\hat{\mathbf{s}} \tag{A7}$$

We consider an isotropic medium and the elasticity tensor $c$ from Eq. (A1) is

$$c_{jkpq} = \delta_{jk}\delta_{pq}\lambda + (\delta_{jp}\delta_{kq} + \delta_{jq}\delta_{kp})\mu \tag{A8}$$

where $\lambda$ and $\mu$ are the Lamé parameters

$$\lambda = \rho(v_P^2 + 2\mu) \qquad \mu = \rho v_S^2 \tag{A9}$$

We set $\lambda = \mu$, resulting in the widely observed relation $v_P = v_S\sqrt{3}$.

  With the assumptions outlined above it is possible to calculate the displacement of an earthquake at location $x$ with 12 parameters (Fig. A1):

－ fault size and orientation: length $L$, width $W$, strike $\phi$, dip $\delta$

－ material: 1st and 2nd Lamé parameters $\lambda$ and $\mu$, density $\rho$ (alternatively: compressional and shear wave velocities $v_P$ and $v_S$ and density $\rho$)

－ rupture and slip: rupture velocity $\zeta$, slip $D_0$, rise time $T$, rake $\lambda$, rupture orientation with respect to strike, $p$

The fault size and displacement of earthquakes are correlated with each other and are scaled to the magnitude. The number of parameters reduces to ten (nine if the Lamé constants are equal), when scaling relations (e.g. Leonard, 2010; Strasser et al.,

2010) are used in combination with the seismic moment $M_0$. The moment can be decomposed in

$$M_0 = \mu A \bar{D} \tag{A10}$$

with shear modulus (2nd Lamé constant) $\mu$, the rupture area—here an ellipse—$A = \frac{\pi}{4}LW$, and average displacement, $\bar{D}$ which follows from Eq. (A4) as $\bar{D} = \frac{2}{3}D_0$.





The results are not strictly comparable to observed data due to the models simplicity. The computed amplitudes will be smaller than observed values, because no free surface is assumed. Assuming a free surface would nearly double the amplitudes from wave reflection, as well as the amplifications from wave transmissions (from high to low velocity zones). Only direct waves are computed, and effects of reflections of different layers are not covered due to the isotropy and homogeneity.

5 Corresponding waveforms—in particular surface waves—are not exhibited. However, the purpose of this model is to show (1) the general behavior of waveforms in the vicinity of a rupture, which is dominated by direct waves, and (2) how amplitudes distribute relatively in space.

*Acknowledgements.* We highly appreciate the help of Tomotaka Iwata and Kimiyuki Asano for providing links to additional seismic data from the municipal and NIED networks and several helpful discussions on the specifics of the data. We are sincerely grateful to Takashi

10 Oguchi, Yuichi Hayakawa, Hitoshi Saito and Yasutaka Haneda for the field trip to the Aso region and fruitful discussions. Thanks to Arno Zang and Odin Marc for various discussions and comments.

Sebastian von Specht, Ugur Ozturk, and Georg Veh acknowledge support from the DFG research training group "Natural Hazards and Risks in a Changing World" (Grant No. GRK 2043/1).




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
