# Peer review of "Effects of finite source rupture on landslide triggering: The 2016 $M_W$ 7.1 Kumamoto earthquake"

_Solid Earth, 2018_

## Referee Comment (RC1) · H. Setiawan (Referee) · 19 Nov 2018

Your manuscript presents the pattern of co-seismic landslides due to the M7.1 Kumamoto earthquake in 2016, related with the ground motion. Since some of the geomorphic parameters did not sufficiently explain the pattern and location of these earthquake-triggered landslides, then the analysis for about 15,000 resulted landslides is carried out by engineering seismology approaches, including rupture directivity effect and the variation of amplitude on a fault-normal and fault-parallel motion. By look at these two key points within a low-frequency ground motion, the spatial pattern distribution and landslide aspects are clearly defined and clarified. In further, results suggest

a physical-based ground-motion model that incorporates the azimuth-dependent seismic energy as well as the moment magnitude. This model would give benefits for co-seismic landslide hazard assessment, due to strike-slip or subduction earthquakes, when the distribution of co-seismic landslides is strongly correlated with the rupture effects rather than the hydrological factors from soil moisture variability or antecedent rainfalls (if any). No significant objections came to this manuscript.

However, some necessary amendments below are needed to clarify: 1. Page 8 line 29, the safety factor of FS<1.5 for unstable hillslopes, is this statement applied for seismic induced, or rainfall-induced or both in general?

2. Page 9 line 7-8, please check the equation of Arias intensity, is it phi/2g or 2/phi.g, see reference i.e Jibson (2007), USGS (1993) or Stafford et al (2009).

3. Page 10 line 6, please add remark M0 for the seismic moment directly. Some equations remarks also should be checked and added (if not yet mentioned).

4. Page 13 line 1, "...since energy is proportional to the seismic moment M0 (Eq.9)..." this should be (Eq.10)?? (Hanks and Kanamori, 1979).

5. Page 13 line 12, "... and 'teta'E and 'teta'E are the azimuths of the maximum." This should be 'teta'E and 'teta'I

6. Figure 14 indicates that mostly landslides concentrated in the aspect of about 120 degrees, south-east, with distance for the rupture approximately within 1-2km, which from location densely surrounding Aso caldera. Besides rupture effects, does distinctive lithology condition in Aso caldera itself also contribute to this finding?

7. Does the rupture propagation energy also (at the end) include the compressional waves (page 9 line 24) in the Aso caldera, south-east side, where the landslides densely concentrated as described in your finding? What is your opinion as an additional explanation in the Discussion? Since your manuscript only applies the shear waves only for estimating the energy in the model.

8. Related to questions 6 and 7, after your findings, do the normal faulting component should be accounted into your model? For example, if we look both at strike-slip and normal components. Does it significantly affect the spatial pattern, asymmetrical distribution or landslides depth?

9. Landslides aspects and asymmetric spatial distribution are well described in your manuscript. Do the depth variability of those recorded co-seismic landslides also can be related with the rupture propagation processes and can be explained through your physical-based ground motion model?

---

## Short Comment (SC1) · 29 Nov 2018

P8 L5-20 : In this paragraph I think you want to cite Marc et al 2016 ( JGR about landslide total area and volume) and not Marc et al 2017 ( about affected area).

P11 : L20 – 25 : the comparison of the 2 equations is a bit misleading because you add 2 terms (anelasic attenuation , and Mw dependent geometric spreading $c_5$), and shuffle the order : $c_3$ becomes $c_4$ in the second equation...
Why not writing the second Eq : $\ln(I) = c_1 + c_2 M + (c_3 + c_4 M)\ln(r) + c_6 r$
Also note that including anelastic attenuation has also been done in various studies ( Meunier 2007, 2013, Yuan 2013) so it is not completely new. Nevertheless, I agree that it is often difficult to constrain the attenuation coefficient and thus, using geometric spreading only is often preferred.

P14 L 11: indicating a landslide failure process starting from the crown and according to simulations by Dang et al. (2016).
Sentence with a missing word ?

Fig 5 : Maybe indicate the Attenuation parameter obtained from the MLE exponetial fit ?

P15: L 1-9: Unclear what is the main message or aim of this paragraph.
 L1 Locations rather than localities ?

L1 "Propagated progressively" : What do you mean? Are you talking about rupture propagation ( as suggested by the following lines ) or about run out ( I.e landslide downslope, rapid, motion after failure). Because unless you specifically restrict analysis to scar areas the "flatter areas" are likely deposit areas, related to runout termination not rupture propagation.

L10 : Mt. Aso and its caldera and Mt. Shutendo‾ji had a high density of landslides (Fig. 5), whereas Mt. Kinpo  and Mt. Otake lack landslides, though these locations are closer to the epicenter and at comparable distances from the rupture (Fig. 5).
Actually it is not so clear on Fig 5 ( Aso is relatively low, similar to Otake). Referring to Fig 7 where the low spatial density is clearer ( and adding Peak Names on this figure) may be better.

Figure 4 : This is a nice ans standard figure.  It may be worth to show the same figure done with the non EQ landslides ? That may be a simple to do supplementary figure that would show nicely if the trend in MAF is significant and due to the EQ ( as there is no reason MAF should relate to rainfall induced landslides).

Fig 6  very nice figure

Fig 8: Density of landslide concentration is an awkward term. I guess it is the Kernel Density estimate of Landslide concentration ( remind the unit of landslide concentration as in following figures)

P20 L2 :"depends on the ratio of rupture and shear wave velocity and the length of the rupture (Spudich and Chiou, 2008)."
Unless rediscussed later , it would be nice to know how :  If fault length and rupture velocity indicates the ptotential importance of directivity for landslide pattern, it may be included in simple models.

Fig 11: On figure 11 I would have like to see the FN/FP in more details for the low frequency range.
 Indeed, how much of the variations in the 0.1-1Hz range remain in the 0.5-1Hz range ? Because we may expect the PGV/PGA at 0.1Hz too weak to cause landsliding, compare to the one between 0.5 and 1 Hz.
If the authors can make it easily I would suggest they split the 0.1-1Hz range in 2 or 3 subplot, as it may yield useful insight for later studies on frequency effects on landslide triggering. Maybe as a supplementary figure, or as a few lines about the contribution of subranges of frequencies.
This is somewhat shown in Fig 10 but as far as I understood Fig 10 is model and Fig 11 is data. The text is somewhat unclear about that and does not call Figure 10, I think.

P25:L32:Maybe not so intractable : Using an estimate of landslide width, and expectations on landslide scar aspect ratio ( Domej et al., 2017)  the scar area can be estimated and the crresponding high elevation pixels can be extracted within each polygons. This requires high quality mapping where individual landslides are not bundled together. But this approach has been validated and shown to improve correlation with rainfall in Marc et al., 2018a, and shown to improve volume nd erosion estimates in Marc et al., 2018b.
This is a side topic for your study, but it could be mentionned, and at least this statement may be more nuanced.

Fig 12 : I would say the caption can be simple and clearer as : a) Aspect and hillslope inclination distribution within areas of the earthquake triggered landslides. This distribution is normalized by the distribution of the aspect of all hillslopes in the landslide affected area

P21 L9 : This is an interesting and important point, but I would maybe rephrase it in terms of slopes. Because it is the slope that control the aspect of a landslide (that is what you measure on your DEM ) and the earthquake is simply preferentially caussing failures in somes slopes ( because wave motions and accelerations are stronger in some specific directions that will increase more or less the slope parallel component leading to failure). So the pattern of ground motion favor landslides in some part of the landscape, and at finer scale the directions of ground motions (FN/FP ratio) will force failure on specific slope aspects.

I would say a a few lines discussing when and how different earthquakes will display strong directivity effect would be a good addition ( maybe starting from your statement about Rupture speed and length ? Cf comment above).

P26 L18 : As commented above, Meunier 2007 (as well as Meunier 2013) consider landslide decay away from the source with a geometric and an exponential decay, similar to anelastic effect.

References used:

Domej, G., Bourdeau, C. and Lenti, L.: Mean Landslide Geometries Inferred from a Global Database of Earthquake- and Non-Earthquake-Triggered Landslides, Italian Journal of Engineering Geology and Environment, (2), 87–107, doi:10.4408/IJEGE.2017-02.O-05, 2017.

Marc, O., Stumpf, A., Malet, J.-P., Gosset, M., Uchida, T. and Chiang, S.-H.: Initial insights from a global database of rainfall-induced landslide inventories: the weak influence of slope and strong influence of total storm rainfall, Earth Surface Dynamics, 6(4), 903–922, doi:https://doi.org/10.5194/esurf-6-903-2018, 2018a.

Marc, O., Behling, R., Andermann, C., Turowski, J. M., Illien, L., Roessner, S. and Hovius, N.: Long-term erosion of the Nepal Himalayas by bedrock landsliding: the role of monsoons, earthquakes and giant landslides, Earth Surface Dynamics Discussions, 1–41, doi:https://doi.org/10.5194/esurf-2018-69, 2018b.

Meunier, P., Uchida, T. and Hovius, N.: Landslide patterns reveal the sources of large earthquakes, Earth and Planetary Science Letters, 363, 27–33, doi:10.1016/j.epsl.2012.12.018, 2013.

Yuan, R.-M., Deng, Q.-H., Cunningham, D., Xu, C., Xu, X.-W. and Chang, C.-P.: Density Distribution of Landslides Triggered by the 2008 Wenchuan Earthquake and their Relationships to Peak Ground Acceleration, Bulletin of the Seismological Society of America, 103(4), 2344–2355, doi:10.1785/0120110233, 2013.

---

## Referee Comment (RC2) · XU (Referee) · 6 Jan 2019

I have read the paper with interest. The authors address a topic that is critical in assessment of spatial distribution of observed slope collapses in the aftermath of strong and moderate earthquakes. For that they have proposed a coherent and to the best of my knowledge, novel approach for explaining the issue at hand. Therefore I recommend publication after minor revision.

1. The unspecified landslide date are from NIED. Please discuss the possible uncertainties may cause by these data. In addition, how accuracy the coseismic inventory is, please also explain the possible mapping errors, and discuss how they will affect the

results.

2. Figure 4: lines are not so visible. Fig.4 b and d are not well explained. Please use a better presentation of the data in this figure.

3. Figure 7: The explanation of this figure in the texts is not enough. Is this point density map? Do you consider the size of the landslides here?

4. Please explain the correlation between the unspecified landslides and the coseismic landslides? Are there any reactivationsïij§

5. Some relevant and important references are missing:

Fan X et al (2018), published on Landslides journal: Coseismic landslides triggered by the 8th August 2017 Ms 7.0 Jiuzhaigou earthquake (Sichuan, China): factors controlling their spatial distribution and implications for the seismogenic blind fault identification.

On Page 3, line 8-9: Wenchuan earthquake has been well studied by many others, please also refer to:

Huang and Fan (2013). "The landslide story" on Nature Geoscience.

---

## Referee Comment (RC3) · Wang (Referee) · 18 Jan 2019

The paper analyzes the seismic effect on seismic landslide due to the propagation of seismic rupture during the 2016 Kumamoto earthquake in central Kyushu (Japan). This is an extremely meticulous and time-consuming process. The results have important implications for the regional seismic landslide development and hazard assessment research. However, there are some details that need further consideration and improvement. (1) In Figure 1 and 5, it is not obvious that Mt. Aso, its caldera, Mt. Shutendoji , Mt. Kinpo and Mt. Otake are near-identical conditions, particularly, the lithology, and topographic characteristics. (2) In Figure 1, it's true that the landslides triggered by

this earthquake are concentrated mainly inside the caldera and the flanks of Mt. Aso. But this area is also nearer the fault rupture patch with highest slip than other three areas. This means more energy could be released from this place during the earthquake. So the difference between distance effect and directivity effect needs to be analyzed. (3) This directivity effect results in larger shaking amplitudes in the rupture propagation direction variations in wave amplitudes and energy related to the directivity effect occur at lower frequencies. The paper shows the total landslide affected area is within 22.9 km distance from the rupture plane. In this near fault area, the effect of high-frequency seismic ground motion on landslide should be more important than the low-frequency. (4) The coseismic landslide is resulted in seismic load and slope geotechnical engineering conditions. This paper mainly makes an in-depth analysis from the engineering earthquake perspective, but the analysis of engineering geological factors is relatively rare. The conclusion is somehow different from some empirical knowledge. I suggest authors further analyze the influence of engineering geological factors. For example, authors can consider the physical and mechanical properties of rock-soil mass and DEM data with higher accuracy to analyze their correlation with landslide, and use quantitative indicators to describe the correlation. These may affect the results to some extent.

Specific comments 1. Figure 1. Add a map scale and identify the epicenter of the Yufu event. 2. The location of mountain peaks should be shown in figure 2a. The details in the four areas listed in figure 5 should be evidenced by zooming in. 3. Page 4. The map scale of the Seamless Digital Geological Map of Japan should be stated. 4. Page4. The computation process of fundamental frequency of hillslope section should be stated. 5. Page 8. Throughout the paper, no coseismic landslide displacement is calculated or used. I suggest delete this part. 6. Page 11. Many empirical attenuation relationships for Arias intensity are developed recent years. Why use the Kramer (1996) model here?

---

## Author Comment (AC1) · 15 Feb 2019

Dear Mr. Setiawan,

thank you for your comments and constructive suggestions, which we considered in detail to improve the presentation of our study. Please find below our point-by-point response to the *original comments*:

1. *page 8, line 29: The safety factor of $F_S < 1.5$ for unstable hillslopes, is this statement applied for seismic induced, or rainfall-induced or both in general?*

   In this particular case we refer to the work by Chen et al. [2017]. They mentioned

both rainfall and earthquakes for their safety factor definition.

Changes in text (p. 9, l. 3):

Chen et al. (2017) characterized unstable hillslopes, related to both rainfall and earthquakes, by a safety factor of $FS < 1.5$. Rarely is the limit equilibrium at $FS = 1$ considered as a reliable metric in engineering geology.

2. *page 9 line 7–8: please check the equation of Arias intensity, is it $\frac{\pi}{2g}$ or $\frac{2}{\pi g}$, see reference i.e Jibson (2007), USGS (1993) or Stafford et al (2009).*

   This was a typo in the fraction indeed.

   Now changed in text (p. 9, l. 14):

   $\frac{\pi}{2g}$

3. *Page 10 line 6: please add remark $M_0$ for the seismic moment directly. Some equations remarks also should be checked and added (if not yet mentioned).*

   Done and changed in text (p. 10, l. 11):

   where $\Delta\sigma$ is the stress drop, $\mu$ the shear modulus, and $M_0$ the seismic moment.

4. *page 13, line 1: "...since energy is proportional to the seismic moment $M_0$ (Eq.9)..." this should be (Eq.10)?? (Hanks and Kanamori, 1979).*

   Yes, changed to Eq. 10. on p. 10, l. 10

5. *Page 13 line 12: "... and $\theta_E$ and $\theta_E$ are the azimuths of the maximum." This should be $\theta_E$ and $\theta_I$*

   Yes, changed to $\theta_I$ on p. 13, l. 29

6. *Figure 14 indicates that mostly landslides concentrated in the aspect of about 120 degrees, south-east, with distance for the rupture approximately within 1– 2 km, which from location densely surrounding Aso caldera. Besides rupture*

*effects, does distinctive lithology condition in Aso caldera itself also contribute to this finding?*

The lithology (or at least the nominal descriptions of dominant rock types) does not show any distinctive directional properties. While it is reasonable to assume that landslides occurred along the weak zones (such as the Halloysite layers we refer to), no preferred orientation has been reported for these shallow layers. (Paudel et al., 2007, 2008; Sato et al., 2017).

7. *Does the rupture propagation energy also (at the end) include the compressional waves (page 9 line 24) in the Aso caldera, south-east side, where the landslides densely concentrated as described in your finding? What is your opinion as an additional explanation in the Discussion? Since your manuscript only applies the shear waves only for estimating the energy in the model.*

The exact calculation of radiated seismic energy is quite complex, which is the main reason why we only consider the shear wave velocity at a site. Our assumption is that we treat the entire waveform as if it arrived at a constant velocity at a site when estimating the radiated seismic energy. This assumption results in an underestimation of the energy of 2.6% at longer distances and 7% at the fault. Compared to other components of the energy estimation procedure, e.g. the assumptions for the geometrical spreading, the usage of the shear wave velocity only introduces a minor error at most sites.

Changed in text (p. 10, l. 11-16):

Since most seismic energy is released as shear waves, we apply the shear wave velocity at the recording site ($v_S$) to the entire waveform, i.e. we assume that all waves arrive with velocity $v_S$ at a site. This assumption has the advantage that it does not require a separation of the record into P- and S-waveforms, simplifying the computation. In the Appendix we show from a theoretical perspective that using a uniform $v_S$ has only a small impact on the overall energy estimate.

We added the detailed description of the appendix.

8. *Related to questions 6 and 7, after your findings, do the normal faulting component should be accounted into your model? For example, if we look both at strike-slip and normal components. Does it significantly affect the spatial pattern, asymmetrical distribution or landslides depth?*

We showed that the fault-normal/fault-parallel ratios are consistent with Somerville et al. (1997), who also formulated their fault-normal/fault-parallel ratio as a term that can be easily plugged into a GMPE. Somerville et al. (1997) also provide model coefficients for fault-normal/fault-parallel ratios for dip-slip events. They observed a similar behaviour for strike-slip and dip-slip events with the dip-slip events exhibiting lesser amplitude variations of fault-normal/fault-parallel ratios and directivity. The formulation of the FN/FP term as an additional (optional) term for GMPEs is common practice (Somerville et al., 1997; Spudich et al., 2004, 2013). Any impact of single components from strike-slip and normal faulting cannot be quantified here as we investigated only a single earthquake. Concerning the landslide depths, please see the next comment.

9. *Landslides aspects and asymmetric spatial distribution are well described in your manuscript. Do the depth variability of those recorded co-seismic landslides also can be related with the rupture propagation processes and can be explained through your physical-based ground motion model?*

Unfortunately, we do not have depth measurements of the landslides. We only know that the coseismic landslides were shallow (Song et al., 2017; Sato et al., 2017; Hung et al., 2017), thus we cannot make a detailed statement about the relation between landslide depth and rupture processes. We could use an empirical scaling between landslide volume and area, but that would introduce additional (and unnecessary) scatter to our model.

**References**

Chi-Wen Chen, Hongey Chen, Lun-Wei Wei, Guan-Wei Lin, Tomoyuki Iida, and Ryuji Yamada. Evaluating the susceptibility of landslide landforms in Japan using slope stability analysis: a case study of the 2016 Kumamoto earthquake. *Landslides*, 14(5):1793–1801, oct 2017. ISSN 1612-510X. doi: 10.1007/s10346-017-0872-1. URL http://link.springer.com/10.1007/s10346-017-0872-1.

Prem P. Paudel, H. Omura, T. Kubota, and T. Inoue. Spatio-temporal patterns of historical shallow landslides in a volcanic area, Mt. Aso, Japan. *Geomorphology*, 88(1-2):21–33, jul 2007. ISSN 0169555X. doi: 10.1016/j.geomorph.2006.10.011. URL http://linkinghub.elsevier.com/retrieve/pii/S0169555X0600465X.

Prem P. Paudel, H. Omura, T. Kubota, and B. Devkota. Characterization of terrain surface and mechanisms of shallow landsliding in upper Kurokawa watershed, Mt Aso, western Japan. *Bulletin of Engineering Geology and the Environment*, 67(1):87–95, feb 2008. ISSN 1435-9529. doi: 10.1007/s10064-007-0108-z. URL http://link.springer.com/10.1007/s10064-007-0108-z.

Tatsuki Sato, Masahiro Chigira, and Yuki Matsushi. Topographic and Geological Features of Landslides Induced by the 2016 Kumamoto Earthquake in the Western Part of the Aso Caldera. *DPRI Annuals*, 60B:431–452, 2017. URL http://hdl.handle.net/2433/229383.

P. G. Somerville, N. F. Smith, R. W. Graves, and N. a. Abrahamson. Modification of Empirical Strong Ground Motion Attenuation Relations to Include the Amplitude and Duration Effects of Rupture Directivity. *Seismological Research Letters*, 68(1):199–222, 1997. ISSN 0895-0695. doi: 10.1785/gssrl.68.1.199.

By Paul Spudich, Brian S J Chiou, Robert Graves, Nancy Collins, and Paul Somerville. A Formulation of Directivity for Earthquake Sources Using Isochrone Theory. *U.S. Geol. Surv. Open-File Rept. 2004-1268*, page 54, 2004.

P. Spudich, J.R. Bayless, J. Baker, Brian S J Chiou, B. Rowshandel, S. Shahi, and Paul Somerville. Final Report of the NGA-West2 Directivity Working Group. Technical Report Final, Pacific Earthquake Engineering Research Center, 2013.

Kun Song, Fawu Wang, Zili Dai, Akinori Iio, Osamu Osaka, and Seiji Sakata. Geological characteristics of landslides triggered by the 2016 Kumamoto earthquake in Mt. Aso volcano, Japan. *Bulletin of Engineering Geology and the Environment*, pages 1–10, jun 2017. ISSN 1435-9529. doi: 10.1007/s10064-017-1097-1. URL http://link.springer.com/10.1007/

s10064-017-1097-1.

Ching Hung, Guan-Wei Lin, Huei-Sian Syu, Chi-Wen Chen, and Hsin-Yi Yen. Analysis of the Aso-Bridge landslide during the 2016 Kumamoto earthquakes in Japan. *Bulletin of Engineering Geology and the Environment*, (200):1–11, jul 2017. ISSN 1435-9529. doi: 10.1007/s10064-017-1103-7. URL http://link.springer.com/10.1007/s10064-017-1103-7.

---

## Author Comment (AC2) · 15 Feb 2019

Dear Mr. Xu,

thank you for your comments and constructive suggestions, which we considered in detail to improve the presentation of our study. Please find below our point-by-point response to the *original comments*:

1. *The unspecified landslide date are from NIED. Please discuss the possible uncertainties may cause by these data. In addition, how accuracy the coseismic inventory is, please also explain the possible mapping errors, and discuss how*

*they will affect the results.*

The landslides were mapped from aerial imagery at much higher resolution (sub-meter resolution) than the 30-m DEM that we used to compute local hillslope aspect and slope. Even a large systematic bias (up to several meters) in the landslide mapping would be small compared to the DEM resolution. We appreciate the reviewer's comment in this regard, but feel that a detailed uncertainty analysis of the landslide inventories is beyond the scope of this study.

2. *Figure 4: lines are not so visible. Fig.4 b and d are not well explained. Please use a better presentation of the data in this figure.*

   The figure has been modified to offer more detailed description and an explanatory figure.

3. *Figure 7: The explanation of this figure in the texts is not enough. Is this point density map? Do you consider the size of the landslides here?*

   Yes, these are kernel density maps of the landslides. The landslide area is taken into account in these figures as highlighted by the colorbar annotation. We added some more detail to the description of Fig. 7.

4. *Please explain the correlation between the unspecified landslides and the coseismic landslides? Are there any reactivations?*

   We discuss the relation between both landslide data sets at the end of the section "Topographic analysis". A detailed study of the unspecified and coseismic landslides is that by Chen et al. (2017), who reported 29 reactivated landslides in the area affected by slope failures. We refer to their work in the text.

   Changed in text (p. 18, l.3):

   Chen et al. (2017) identified only 29 landslide recactivations during the Kumamoto earthquake.

5. *Some relevant and important references are missing: Fan X et al (2018), published on Landslides journal: Coseismic landslides triggered by the 8th August 2017 Ms 7.0 Jiuzhaigou earthquake (Sichuan, China): factors controlling their spatial distribution and implications for the seismogenic blind fault identification.*

   We added the suggested reference to the introduction, page 1 line 23 and page 3 line 10.

6. *On Page 3, line 8-9: Wenchuan earthquake has been well studied by many others, please also refer to: Huang and Fan (2013). "The landslide story" on Nature Geoscience.*

   We agree with the reviewer: literally hundreds of papers on the Wenchuan earthquake have become available by now. Yet the suggested reference cites papers that we already cite concerning the earthquake related landslides of Wenchuan, i.e. Gorum et al. (2011).

**References**

Chi-Wen Chen, Hongey Chen, Lun-Wei Wei, Guan-Wei Lin, Tomoyuki Iida, and Ryuji Yamada. Evaluating the susceptibility of landslide landforms in Japan using slope stability analysis: a case study of the 2016 Kumamoto earthquake. *Landslides*, 14(5):1793–1801, oct 2017. ISSN 1612-510X. doi: 10.1007/s10346-017-0872-1. URL http://link.springer.com/10.1007/s10346-017-0872-1.

Tolga Gorum, Xuanmei Fan, Cees J. van Westen, Run Qiu Huang, Qiang Xu, Chuan Tang, and Gonghui Wang. Distribution pattern of earthquake-induced landslides triggered by the 12 May 2008 Wenchuan earthquake. *Geomorphology*, 133(3-4):152–167, 2011. ISSN 0169555X. doi: 10.1016/j.geomorph.2010.12.030. URL http://dx.doi.org/10.1016/j.geomorph.2010.12.030.

---

## Author Comment (AC3) · 15 Feb 2019

Dear Mr . Wang,

thank you for your comments and constructive suggestions, which we considered in detail to improve the presentation of our study. Please find below our point-by-point response to the *original comments*:

1. *In Figure 1 and 5, it is not obvious that Mt. Aso, its caldera, Mt. Shutendoji, Mt. Kinpo and Mt. Otake are near-identical conditions, particularly, the lithology, and topographic characteristics.*

[Figure]

We used the term "near-identical condition" to outline that the four mountains have all geological young volcanic rocks of similar composition, and that hillslope inclination and MAF are elevated.

2. *In Figure 1, it's true that the landslides triggered by this earthquake are concentrated mainly inside the caldera and the flanks of Mt. Aso. But this area is also nearer the fault rupture patch with highest slip than other three areas. This means more energy could be released from this place during the earthquake. So the difference between distance effect and directivity effect needs to be analyzed.*

We agree on the statement that energy release is localized in the asperity. We address this by considering the asperity portion only and show the landslide distribution with asperity distance in a new figure (Fig. 5b) and landslide azimuth with respect to the asperity centroid (Fig 6b). Given the extent and steepness of the asperity patch, results change slighlty when compared to the results for the entire rupture plane.

3. *This directivity effect results in larger shaking amplitudes in the rupture propagation direction variations in wave amplitudes and energy related to the directivity effect occur at lower frequencies. The paper shows the total landslide affected area is within 22.9 km distance from the rupture plane. In this near fault area, the effect of high-frequency seismic ground motion on landslide should be more important than the low-frequency.*

We partly agree on these statements, and see that some clarification is needed. We never stated that the lower-frequency contribution is more important; instead we say that it considerably contributes to the overall shaking and landslide triggering. The majority of landslides has aspects that cannot be explained solely by the lower-frequency ground motion and its associated directivity (Fig. 14). Because of these observations, we base our GMPE on Arias intensity. We also stated at the end of section 5.2 that the Arias intensity is more sensitive towards higher

frequency contributions and that it explains most of the data, but not all of them. The ground motion contributions of the lower frequencies—related to energy—is lower than the ground motions at higher frequencies, as shown by our model. Adding an energy term helps to better explain ground motion, though it does by no means explain the entire ground motion. Considering the comments by the reviewer, we recognize that some of our statements are ambiguous. Hence, we made clearer statements in the conclusions.

Changes in text (in page 27, lines 5-6, lines 13-15, lines 22-23.):

We demonstrated that the pattern of coseismic landslides is not only consistent with ground motion at higher frequencies (e.g. distance dependence) but also contributions from lower frequencies are evident.

We introduced a modified model for Arias intensity using site-dependent seismic energy estimates instead of the source-dependent seismic magnitude to better model low-frequency ground-motion in addition to the ground-motion at higher frequencies covered by the Arias intensity.

4. *The coseismic landslide is resulted in seismic load and slope geotechnical engineering conditions. This paper mainly makes an in-depth analysis from the engineering earthquake perspective, but the analysis of engineering geological factors is relatively rare. The conclusion is somehow different from some empirical knowledge. I suggest authors further analyze the influence of engineering geological factors. For example, authors can consider the physical and mechanical properties of rock-soil mass and DEM data with higher accuracy to analyze their correlation with landslide, and use quantitative indicators to describe the correlation. These may affect the results to some extent.*

Our work focuses on the seismological part and less on the geological aspects, as these have been analyzed in detail by others for the Kumamoto region in context of the 2016 earthquake. Analysis of physical and mechanical properties of

rock-soil mass has been conducted by several other authors (Dang et al., 2016; Song et al., 2017; Paudel et al., 2007, 2008; Sato et al., 2017) and we refer to those works accordingly. Including analyses concerning geotechnical factors at the same scale would expand the entire paper considerably, and is beyond the scope of a single publication and beyond our original objective. All metrics and methods are derived from accepted works in both geotechnical engineering and engineering seismology (e.g Harp and Wilson, 1995; Somerville et al., 1997). One of our key results—the influence of rupture directivity on landslide patterns— has been speculated about in previous studies (e.g. Hovius and Meunier, 2012). We also fail to see direct benefits of using a DEM of higher resolution and performing geotechnical analyses for the regional pattern of landsliding that we are interested in explaining. In any case, we use the DEM with highest resolution that is freely available for the region.

5. *Figure 1. Add a map scale and identify the epicenter of the Yufu event.*

   Like in the other maps, the map scale is now given in form of UTM coordinates and the event epicenter has been added.

6. *The location of mountain peaks should be shown in figure 2a. The details in the four areas listed in figure 5 should be evidenced by zooming in.*

   The mountain peaks haven been added to Fig. 2.

7. *Page 4. The map scale of the Seamless Digital Geological Map of Japan should be stated.*

   Yes, done.

   Changes in text (page 4, line 11-12):

   While data on major geological units are from the Seamless Digital Geological Map of Japan (scale 1:200,000) by the Geological Survey of Japan.

8.  *Page 4. The computation process of fundamental frequency of hillslope section should be stated.*

    We do not compute the fundamental frequency; this is not necessary for the purpose of the computation of the median amplification factor (MAF). To avoid further confusion with the fundamental frequency of the hillslope, we deleted the following sentence: "The frequency $f$ of the seismic wave is the fundamental frequency of the hillslope section on which landsliding occurred (Massa et al., 2014)." We see that this sentence might imply that MAF requires knowledge of the fundamental frequency of the hillslope. The frequency $f$ as it used for the computation of MAF, is the frequency of the seismic wave.

9.  *Page 8. Throughout the paper, no coseismic landslide displacement is calculated or used. I suggest delete this part.*

    We use the coseismic landslide displacement relation to show that it is related to acceleration and velocity, as our presented GMPE does. We clarified its purpose.

    Changes in text (in page 9 lines 7-9):

    Thus, the coseismic hillslope performance can be characterized by velocity and acceleration. In the following sections, we derive a ground-motion model based on the acceleration related Arias intensity and the velocity related radiated seismic energy.

10. *Page 11. Many empirical attenuation relationships for Arias intensity are developed recent years. Why use the Kramer (1996) model here?*

    As stated in the text, the functional form of Kramer (1996) is a template, and most ground motion prediction equations—including most recent ones—are related to it. We use the Kramer (1996) template function to highlight that our functional model does not differ from the bulk of other GMPEs. We clarified this and rewrote the first paragraph of the section related to the landslide related ground-motion models (page 13 lines 8-15).

**References**

Khang Dang, Kyoji Sassa, Hiroshi Fukuoka, Naoki Sakai, Yuji Sato, Kaoru Takara, Lam Huu Quang, Doan Huy Loi, Pham Van Tien, and Nguyen Duc Ha. Mechanism of two rapid and long-runout landslides in the 16 April 2016 Kumamoto earthquake using a ring-shear apparatus and computer simulation (LS-RAPID). *Landslides*, 13(6):1525–1534, dec 2016. ISSN 1612-510X. doi: 10.1007/s10346-016-0748-9. URL http://dx.doi.org/10.1007/s10346-016-0748-9http://link.springer.com/10.1007/s10346-016-0748-9.

Kun Song, Fawu Wang, Zili Dai, Akinori Iio, Osamu Osaka, and Seiji Sakata. Geological characteristics of landslides triggered by the 2016 Kumamoto earthquake in Mt. Aso volcano, Japan. *Bulletin of Engineering Geology and the Environment*, pages 1–10, jun 2017. ISSN 1435-9529. doi: 10.1007/s10064-017-1097-1. URL http://link.springer.com/10.1007/s10064-017-1097-1.

Prem P. Paudel, H. Omura, T. Kubota, and T. Inoue. Spatio-temporal patterns of historical shallow landslides in a volcanic area, Mt. Aso, Japan. *Geomorphology*, 88(1-2):21–33, jul 2007. ISSN 0169555X. doi: 10.1016/j.geomorph.2006.10.011. URL http://linkinghub.elsevier.com/retrieve/pii/S0169555X0600465X.

Prem P. Paudel, H. Omura, T. Kubota, and B. Devkota. Characterization of terrain surface and mechanisms of shallow landsliding in upper Kurokawa watershed, Mt Aso, western Japan. *Bulletin of Engineering Geology and the Environment*, 67(1):87–95, feb 2008. ISSN 1435-9529. doi: 10.1007/s10064-007-0108-z. URL http://link.springer.com/10.1007/s10064-007-0108-z.

Tatsuki Sato, Masahiro Chigira, and Yuki Matsushi. Topographic and Geological Features of Landslides Induced by the 2016 Kumamoto Earthquake in the Western Part of the Aso Caldera. *DPRI Annuals*, 60B:431–452, 2017. URL http://hdl.handle.net/2433/229383.

Edwin L Harp and Raymond C Wilson. Shaking Intensity Thresholds for Rock Falls and Slides : Evidence from 1987 Whittier Narrows and Superstition Hills Earthquake Strong-Motion Records. *Bulletin of the Seismological Society of America*, 85(6):1739–1757, 1995.

P. G. Somerville, N. F. Smith, R. W. Graves, and N. a. Abrahamson. Modification of Empirical Strong Ground Motion Attenuation Relations to Include the Amplitude and Duration Effects of Rupture Directivity. *Seismological Research Letters*, 68(1):199–222, 1997. ISSN 0895-0695. doi: 10.1785/gssrl.68.1.199.

Niels Hovius and Patrick Meunier. Earthquake ground motion and patterns of seismically induced landsliding. In John J. Clague and Douglas Stead, editors, *Landslides*, pages 24–36. Cambridge University Press, Cambridge, 2012. doi: 10.1017/CBO9780511740367.004. URL https://www.cambridge.org/core/product/identifier/9780511740367{%}23c00206-3-1/type/book{_}part.

Marco Massa, Simone Barani, and Sara Lovati. Overview of topographic effects based on experimental observations: meaning, causes and possible interpretations. *Geophysical Journal International*, 197(3):1537–1550, jun 2014. ISSN 0956-540X. doi: 10.1093/gji/ggt341. URL https://academic.oup.com/gji/article-lookup/doi/10.1093/gji/ggt341.

Steven L Kramer. Geotechnical earthquake engineering. In prentice–Hall international series in civil engineering and engineering mechanics. *Prentice-Hall, New Jersey*, 1996.

---

## Author Comment (AC4) · 15 Feb 2019

Dear Odin,

thank you for commenting on our manuscript. We considered your suggestions in detail to improve the presentation of our study. Please find below our point-by-point response to the *original comments*:

1. *P8 L5-20 : In this paragraph I think you want to cite Marc et al 2016 ( JGR about landslide total area and volume) and not Marc et al 2017 ( about affected area).*

   Reference is corrected.

[Figure]

2. *P11 : L20 – 25 : the comparison of the 2 equations is a bit misleading because you add 2 terms (anelasic attenuation ,and Mw dependent geometric spreading c5), and shuffle the order : c3 becomes c4 in the second equation... Why not writing the second Eq : ln(I) = c1 + c2M + (c3+c4M)ln(r) + c6 r Also note that including anelastic attenuation has also been done in various studies ( Meunier 2007, 2013, Yuan 2013) so it is not completely new. Nevertheless, I agree that it is often difficult to constrain the attenuation coefficient and thus, using geometric spreading only is often preferred.*

The entire equation paragraph has been rephrased in more detail and the first equation uses now coefficients $p_i$ that are compared and discussed with coefficients $c_j$ of the second equation. The remarks on anelastic attenuation are now incorporated (page 13 lines 8-15). This refers also to the last comment.

3. *P14 L 11: indicating a landslide failure process starting from the crown and according to simulations by Dang et al. (2016). Sentence with a missing word ?*

Indeed: indicating a landslide failure process starting from the crown and which is according to simulations by Dang et al. (2016).

4. *Fig 5 : Maybe indicate the Attenuation parameter obtained from the MLE exponential fit ?*

This comment is in connection to the comment for Fig. 9. The y-axis label reads now landslide concentration and is consistent now with Fig. 9, 12, 14, and 15. The figure has been reworked to account also for different distance metrics (rupture distance and asperity distance). The parameter estimates have been added to the figure.

5. *P15: L 1-9: Unclear what is the main message or aim of this paragraph. L1 Locations rather than localities ? L1 "Propagated progressively" : What do you mean? Are you talking about rupture propagation ( as suggested by the following*

*lines ) or about run out ( I.e landslide downslope, rapid, motion after failure). Because unless you specifically restrict analysis to scar areas the "flatter areas" are likely deposit areas, related to runout termination not rupture propagation.*

This paragraph is specifically about the rupture process.

The first sentence has been changed as follows in page 15 line 3:

Most landslides originated at locations with amplified ground accelerations and steep hillslopes and propagated progressively to flatter areas with less amplified ground accelerations and deposited the material in areas of attenuated ground accelerations.

6. *L10 : Mt. Aso and its caldera and Mt. Shutendoji had a high density of landslides (Fig. 5), whereas Mt. Kinpo and Mt. Otake lack landslides, though these locations are closer to the epicenter and at comparable distances from the rupture (Fig. 5). Actually it is not so clear on Fig 5 ( Aso is relatively low, similar to Otake). Referring to Fig 7 where the low spatial density is clearer ( and adding Peak Names on this figure) may be better.*

Another reviewer made similar remarks and the figures showing the peaks have been updated.

7. *Figure 4 : This is a nice ans standard figure. It may be worth to show the same figure done with the non EQ landslides ? That may be a simple to do supplementary figure that would show nicely if the trend in MAF is significant and due to the EQ ( as there is no reason MAF should relate to rainfall induced landslides).*

The NIED landslide inventory is unspecified with regard to the landslide trigger and we cannot extract the non-seismic landslides.

8. *Fig 6 very nice figure*

Thanks!

9. *Fig 8: Density of landslide concentration is an awkward term. I guess it is the Kernel Density estimate of Landslide concentration ( remind the unit of landslide concentration as in following figures)*

In Figure 9 (and Fig. 5 and 15 for that matter) the density is normalized to integrate to 1, while in figures 12 and 14 it scales to the landslide concentration, hence the different labels and units. Figure 5, 9 and 15 have been updated and the labeling and scaling is consistent with figures 12 and 14 with landslide concentration. This has implications on the parameters in Eq. 27 and 28 which have been accordingly updated.

10. *P20 L2 :"depends on the ratio of rupture and shear wave velocity and the length of the rupture (Spudich and Chiou, 2008)." Unless rediscussed later, it would be nice to know how: If fault length and rupture velocity indicates the ptotential importance of directivity for landslide pattern, it may be included in simple models.*

The work of Spudich and Chiou (2008) explains in detail the procedure on how to use these parameters. Unfortunately, even their simplified model is relatively complex as it needs to account for the seismic radiation pattern. Describing it in more detail in this paper while not using the method would go too much off-topic and we have investigated one earthquake only here. Like the Somerville et al. (1997) model, it can be added as a term to existing GMPEs.

11. *Fig 11: On figure 11 I would have like to see the FN/FP in more details for the low frequency range. Indeed, how much of the variations in the 0.1-1Hz range remain in the 0.5-1Hz range ? Because we may expect the PGV/PGA at 0.1Hz too weak to cause landsliding, compare to the one between 0.5 and 1 Hz. If the authors can make it easily I would suggest they split the 0.1-1Hz range in 2 or 3 subplot, as it may yield useful insight for later studies on frequency effects on landslide triggering. Maybe as a supplementary figure, or as a few lines about the contribution of subranges of frequencies. This is somewhat shown in Fig 10 but*

*as far as I understood Fig 10 is model and Fig 11 is data. The text is somewhat unclear about that and does not call Figure 10, I think.*

This comment may arise due to the missing link between Fig. 10 and 11. and the requested frequency dependence is actually adressed by Fig. 10. Similarly to Fig. 11, we added now the KDE in Fig. 10 to show the data distribution. In the text, we link Fig. 10 and Fig. 11 better.

12. *P25:L32: Maybe not so intractable : Using an estimate of landslide width, and expectations on landslide scar aspect ratio ( Domej et al., 2017) the scar area can be estimated and the crresponding high elevation pixels can be extracted within each polygons. This requires high quality mapping where individual landslides are not bundled together. But this approach has been validated and shown to improve correlation with rainfall in Marc et al., 2018a, and shown to improve volume nd erosion estimates in Marc et al., 2018b. This is a side topic for your study, but it could be mentionned, and at least this statement may be more nuanced.*

This comment highlights the landslide complexity and we changed the sentence.

Changes in text:

The reconstruction of the landslide failure planes is limited to statistical assessments of landslide inventories (Domej et al., 2017; Marc et al., 2019).

13. *Fig 12 : I would say the caption can be simple and clearer as : a) Aspect and hillslope inclination distribution within areas of the earthquake triggered landslides. This distribution is normalized by the distribution of the aspect of all hillslopes in the landslide affected area.*

The formulation is less redundant and we changed it as stated.

14. *P21 L9 : This is an interesting and important point, but I would maybe rephrase it in terms of slopes. Because it is the slope that control the aspect of a landslide (that is what you measure on your DEM ) and the earthquake is simply*

*preferentially caussing failures in somes slopes ( because wave motions and ac-
celerations are stronger in some specific directions that will increase more or less
the slope parallel component leading to failure). So the pattern of ground motion
favor landslides in some part of the landscape, and at finer scale the directions
of ground motions (FN/FP ratio) will force failure on specific slope aspects. I
would say a a few lines discussing when and how different earthquakes will dis-
play strong directivity effect would be a good addition ( maybe starting from your
statement about Rupture speed and length ? Cf comment above).*

We made changes and rephrased it in terms of slopes. As mentioned before,
we investigated only one earthquake here. While the directivity effect is well
studied and basically all major earhquakes display it, deriving generalizations
(e.g. behaviour as function of rupture speed) with regard to interactions with
landslide requires the investigation of more earthquake data. At this point, we
could only speculate for the general case.

Changes in text:

This highlights that the earthquake affects the landslide locations (Fig. 6, 7); and
will force failure on specific slopes facing in the direction of ground motion (Fig.
12, 14).

15. *P26 L18 : As commented above, Meunier 2007 (as well as Meunier 2013) con-
sider landslide decay away from the source with a geometric and an exponential
decay, similar to anelastic effect.*

The highlighted sentence is formulated in an ambiguous way. It should read
that Meunier et al. (2007)—as one of few—incorporates the attenuation term in
landslide related GMMs.

Change in text:

The latter is commonly not incorporated in landslide studies but has been incor-
porated by more recent studies (e.g. Meunier et al., 2007; Massey et al., 2018).

**References**

Khang Dang, Kyoji Sassa, Hiroshi Fukuoka, Naoki Sakai, Yuji Sato, Kaoru Takara, Lam Huu Quang, Doan Huy Loi, Pham Van Tien, and Nguyen Duc Ha. Mechanism of two rapid and long-runout landslides in the 16 April 2016 Kumamoto earthquake using a ring-shear apparatus and computer simulation (LS-RAPID). *Landslides*, 13(6):1525–1534, dec 2016. ISSN 1612-510X. doi: 10.1007/s10346-016-0748-9. URL http://dx.doi.org/10.1007/s10346-016-0748-9http://link.springer.com/10.1007/s10346-016-0748-9.

Paul Spudich and Brian S J Chiou. Directivity in NGA earthquake ground motions: Analysis using isochrone theory. *Earthquake Spectra*, 24(1):279–298, 2008. ISSN 87552930. doi: 10.1193/1.2928225.

P. G. Somerville, N. F. Smith, R. W. Graves, and N. a. Abrahamson. Modification of Empirical Strong Ground Motion Attenuation Relations to Include the Amplitude and Duration Effects of Rupture Directivity. *Seismological Research Letters*, 68(1):199–222, 1997. ISSN 0895-0695. doi: 10.1785/gssrl.68.1.199.

G Domej, C Bourdeau, L Lenti, S Martino, and K Pluta. Mean landslide geometries inferred from globla database. *94 Italian Journal of Engineering Geology and Environment*, 2:87–107, 2017. doi: 10.4408/IJEGE.2017-02.O-05.

Odin Marc, Robert Behling, Christoff Andermann, Jens M Turowski, Luc Illien, Sigrid Roessner, and Niels Hovius. Long-term erosion of the Nepal Himalayas by bedrock landsliding: the role of monsoons, earthquakes and giant landslides. *Earth Surface Dynamics*, 7(1):107–128, jan 2019. ISSN 2196-632X. doi: 10.5194/esurf-7-107-2019. URL https://www.earth-surf-dynam.net/7/107/2019/.

Patrick Meunier, Niels Hovius, and A. John Haines. Regional patterns of earthquake-triggered landslides and their relation to ground motion. *Geophysical Research Letters*, 34(20):1–5, 2007. ISSN 00948276. doi: 10.1029/2007GL031337.

C. Massey, D. Townsend, E. Rathje, K. E. Allstadt, B. Lukovic, Y. Kaneko, B. Bradley, J. Wartman, R. W. Jibson, D. N. Petley, N. Horspool, I. Hamling, J. Carey, S. Cox, J. Davidson, S. Dellow, J. W. Godt, C. Holden, K. Jones, A. Kaiser, M. Little, B. Lyndsell, S. McColl, R. Morgenstern, F. K. Rengers, D. Rhoades, B. Rosser, D. Strong, C. Singeisen, and M. Villeneuve. Landslides Triggered by the 14 November 2016 Mw 7.8 Kaikoura Earthquake, New Zealand. *Bulletin of the Seismological Society of America*, 108(3B):1630–1648, jul 2018. ISSN 0037-1106. doi: 10.1785/0120170305. URL https://pubs.geoscienceworld.org/

ssa/bssa/article/529881/Landslides-Triggered-by-the-14-November-2016-Mw78https:
//pubs.geoscienceworld.org/ssa/bssa/article/108/3B/1630/529881/
Landslides-Triggered-by-the-14-November-2016-Mw78.